**DOI: 10.1038/ncomms15086**　　**OPEN**

# Circulating tumour DNA sequence analysis as an alternative to multiple myeloma bone marrow aspirates

Olena Kis[1], Rayan Kaedbey[1], Signy Chow[1,2], Arnavaz Danesh[1], Mark Dowar[1], Tiantian Li[1], Zhihua Li[1], Jessica Liu[1], Mark Mansour[1], Esther Masih-Khan[1], Tong Zhang[1], Scott V. Bratman[1,2], Amit M. Oza[1], Suzanne Kamel-Reid[1,2], Suzanne Trudel[1,2] & Trevor J. Pugh[1,2]

The requirement for bone-marrow aspirates for genomic profiling of multiple myeloma poses an obstacle to enrolment and retention of patients in clinical trials. We evaluated whether circulating cell-free DNA (cfDNA) analysis is comparable to molecular profiling of myeloma using bone-marrow tumour cells. We report here a hybrid-capture-based Liquid Biopsy Sequencing (LB-Seq) method used to sequence all protein-coding exons of *KRAS*, *NRAS*, *BRAF*, *EGFR* and *PIK3CA* in 64 cfDNA specimens from 53 myeloma patients to >20,000 × median coverage. This method includes a variant filtering algorithm that enables detection of tumour-derived fragments present in cfDNA at allele frequencies as low as 0.25% (median 3.2%, range 0.25–46%). Using LB-Seq analysis of 48 cfDNA specimens with matched bone-marrow data, we detect 49/51 likely somatic mutations, with subclonal hierarchies reflecting tumour profiling (96% concordance), and four additional mutations likely missed by bone-marrow testing (>98% specificity). Overall, LB-Seq is a high fidelity adjunct to genetic profiling of bone-marrow in multiple myeloma.

[1] Princess Margaret Cancer Centre, University Health Network, Toronto, Ontario M5G 1L7, Canada. [2] Department of Medical Biophysics, University of Toronto, Toronto, Ontario M5G 1L7, Canada. Correspondence and requests for materials should be addressed to S.T. (email: suzanne.trudel@uhn.ca) or to T.J.P. (email: trevor.pugh@utoronto.ca).

Multiple myeloma (MM) is characterized by recurrent cytogenetic and molecular abnormalities including translocations of the immunoglobulin heavy chain (IgH) locus; chromosomal trisomies; partial deletions or monosomies of chromosomes 1, 13 and 17; and somatic mutations in genes encoding proteins belonging to several signalling pathways[1–4]. Activating mutations in *KRAS*, *NRAS* and *BRAF* genes that encode proteins with a key role in the mitogen-activated protein kinase (MAPK) pathway have been reported in 23–26%, 20–24% and 4–6% of MM cases, respectively[3,4]. Several therapies have been developed to target this pathway in several human malignancies including MM[5,6], and the presence of mutations in these genes is emerging as a requirement for enrolment on new clinical trials[7,8].

In MM, isolation of tumour DNA requires collection of bone marrow (BM) aspirates using a large gauge needle to penetrate muscle and bone that may be difficult, painful and associated with significant patient anxiety as well as rare but significant complications including bleeding and infection. To enrich for malignant plasma cells, CD138 + cells are isolated by flow cytometry or antibody-coated magnetic microbeads[9]. In some cases, DNA extracted from enriched samples cannot be used for genetic profiling due to suboptimal tumour content or low cellular yield[10]. In other cases, the BM may be packed with malignant cells and result in a dry tap (failure to obtain an aspirate). In addition, MM consists of multiple tumours with substantial clonal heterogeneity that infiltrate many of the BM-containing bones in a non-confluent manner[11]. Thus, molecular profiling of a single BM biopsy site may inadequately represent complete tumour burden.

Previous studies have demonstrated that cell-free DNA (cfDNA) isolated from blood plasma of cancer patients contains tumour-derived DNA fragments that are shed into the bloodstream by cancer cells[12,13]. Hybrid-capture and ultra-deep targeted sequencing of mutation hotspots using cfDNA has been described in solid tumours[12,14], but alternative modalities to access haematological cancer genomes in a more unbiased way have been largely overlooked. We therefore set out to investigate whether sequence analysis of cfDNA may be used to assess mutations in MM at a level comparable to clinical laboratory testing of BM aspirates.

In this study, we develop and validate a hybrid-capture-based Liquid Biopsy Sequencing (LB-Seq) method for targeted deep sequencing of all protein coding exons of *KRAS*, *NRAS*, *BRAF*, *EGFR* and *PIK3CA* genes in 64 cfDNA specimens from 53 MM patients. We demonstrate that cfDNA sequence analysis in MM is feasible and can accurately detect 96% of mutations identified by genetic profiling of matched BM-derived tumour DNA with >98% specificity, similar subclonal hierarchies and good concordance between serial plasma samples. This method can potentially replace medically unnecessary BM sampling and provide an alternative non-invasive test for longitudinal genetic monitoring of MM patients receiving targeted therapy.

## Results

**BM aspirate yields**. For algorithm training and validation, we compiled a list of somatic mutations that were detected by genetic profiling of the matching BM aspirates. In total, seven of 48 BM samples collected had ≤1% yield of BM-derived malignant plasma cells (Supplementary Data 1). Thus, 15% of the BM samples were unsuitable for clinical testing, consistent with a previous MM gene expression study[10]. For three BM aspirate samples unsuitable for clinical grade testing due to low yield of malignant plasma cells, we were able to obtain reliable sequencing data using our 5-gene targeted deep sequencing assay because of

lower requirement for input material ($\sim$100 ng of purified tumour DNA) and much higher depth of coverage ($>$5,000 $\times$) compared to clinical testing ($\sim$500 $\times$). When BM sequencing data were available from more than one source (Supplementary Table 1), we used data generated using our 5-gene panel.

**Cell-free DNA yields**. We processed 64 plasma samples from 53 MM patients (11 newly diagnosed and 42 relapsed, of which 13 were enrolled onto PHL-9460, a clinical trial of trametinib) who received a median of three prior lines of therapy (range 0–9, Table 1). We detected higher cfDNA concentrations in this MM cohort compared to 56 patients with advanced solid tumours also processed in our laboratory using the same method (median 20.1 versus 10.3 ng per ml plasma, *P* value < 0.001, Fig. 1a. See Supplementary Table 2 for cancer types). These high cfDNA yields may not be reflective of patients with earlier stage myeloma. High inter-patient variability in cfDNA yields among patients with MM (4.7–331 ng per ml plasma) and other malignancies (0–365 ng per ml plasma) observed in this study was consistent with other studies of patients with advanced cancers[15]. Concentrations of cfDNA in MM patients were independent of the international staging system prognostic index (Fig. 1b), but correlated with advanced disease (late relapse compared to early relapse; *P* value 0.016, Fig. 1c) and appeared higher (not significant) in patients with above normal ($>$220 U l$^{-1}$) lactate dehydrogenase concentrations, a biomarker of tumour burden, proliferation and extramedullary disease (Fig. 1d). Other clinical data collected for 53 MM patients enrolled in the study are available in Supplementary Data 1. Using a binomial sampling model, we predicted that an aliquot of 83 ng of cfDNA has a 99.99% chance to contain a tumour-derived DNA fragment within a population of 2,000 cfDNA fragments in plasma (that is, 0.05%, Fig. 2). Hence, we used 83 ng of cfDNA or less (13 samples had 10–80 ng of cfDNA available), extracted from 3 to 13 ml (median 8.5 ml) of blood plasma, as input for LB-Seq library construction, target capture and sequencing (Supplementary Methods, Supplementary Figs 1–5 and Supplementary Table 3). As described in detail in the Supplementary Methods section, we estimated that for samples with 83 ng of cfDNA used to prepare sequencing libraries (approximately $4.3 \times 10^{11}$ unique cfDNA fragments), a median 43% ($1.8 \times 10^{11}$) of fragments were retained following adapter-ligation, library amplification and bead clean up. Owing to having a small target region (17.5 kb), only a small fraction of these molecules correspond to the region of interest, providing an estimated $1.1 \times 10^6$ unique cfDNA fragments available for capture. This corresponds to unique cfDNA fragment coverage of approximately 11,000 $\times$ for a region with 20,000 $\times$ coverage ($\sim$55%). This percentage will decrease in regions with higher coverage ($\sim$11% unique reads for regions with 100,000 $\times$ coverage). Although target capture is also associated with additional loss of library fragments due to the inefficient hybridization of target probes to their target DNA fragments and loss of hybridized DNA during streptavidin bead capture, the redundancy of library molecules available for capture should prevent significant reduction in library complexity during target capture process. In this study, we were unable to measure the efficiency of target capture; however, a previous study by Newman *et al.* estimated using molecular barcoding and mark and recapture method that approximately 50–60% of unique cfDNA fragments that enter into the library preparation process are represented in the final sequencing data after target capture and PCR[16].

**Training and validation of LB-Seq mutation detection**. Somatic mutation detection in cfDNA is challenged by the presence of base substitution artefacts introduced during library construction and Illumina sequencing, as well as the need to distinguish

**Table 1 | Patient demographics and disease characteristics.**

| Characteristic | Study population ($n = 53$) |
|---|---|
| *MM subtype and FLC type, number (%)* | |
| IgG kappa | 21 (39.6) |
| IgG lambda | 6 (11.3) |
| IgA kappa | 5 (9.4) |
| IgA lambda | 5 (9.4) |
| Light chain kappa | 7 (13.2) |
| Light chain lambda | 5 (9.4) |
| IgD kappa | 1 (1.9) |
| Non-secretory myeloma | 1 (1.9) |
| Oligosecretory myeloma | 1 (1.9) |
| Unknown | 1 (1.9) |
| | |
| *ISS stage at diagnosis, number (%)* | |
| Stage 1 | 14 (26.4) |
| Stage 2 | 8 (15.1) |
| Stage 3 | 17 (32.1) |
| Unknown | 14 (26.4) |
| | |
| Number of previous therapies, median (range) | 3 (0–9) |
| | |
| *Types of previous therapies, number (%)* | |
| PI therapy | 37 (69.8) |
| IMiD therapy | 37 (69.8) |
| ASCT | 32 (60.4) |
| | |
| *Disease status at blood draw\*, number (%)* | |
| Newly diagnosed | 11 (17.2) |
| Early relapsed (1–3 prior lines of therapy) | 28 (43.8) |
| Late relapsed (> 3 prior lines of therapy) | 25 (39.1) |
| | |
| LDH at blood draw†, median (range), U l$^{-1}$ | 261 (140–1,303) |
| M spike at blood draw‡, median (range), g l$^{-1}$ | 21.5 (0.2–84) |
| | |
| *Light chain ratio, median (range)* | |
| kappa subtype: kappa/lambda ratio | 376 (2.0–18,723) |
| lambda subtype: lambda/kappa ratio | 241 (2.2–23,679) |
| | |
| Percentage of plasma cells in BM aspirate#, median (range) | 11 (0.2–80) |

ASCT, autologous stem cell transplantation; IMiD, immunomodulatory drug; ISS, international staging system; LDH, lactate dehydrogenase; M spike, monoclonal immunoglobulin level; PI, protease inhibitor.
*Data set included 64 blood samples collected from 53 patients with nine patients providing two samples each and one patient providing three samples.
†Data available for 52 of 64 plasma samples.
‡Data available for 44 of 64 plasma samples.
#Data available for 41 BM samples from 34 patients.
Additional patient demographics and disease characteristics are provided in Supplementary Data 1.

germline variants from true somatic mutations[17]. To optimize the algorithm for identifying likely somatic mutations in cfDNA sequencing data (Supplementary Figs 6 and 7), we defined a training data set of 25 cfDNA samples from 23 patients that had matching BM-derived tumour DNA. We then validated our method using a blinded sequential validation cohort of 19 cfDNA samples from 17 patients with matching BM sequencing data (Supplementary Fig. 8). Four additional samples included in the validation set were serial cfDNA specimens from patients included in the training cohort (Fig. 3,\*) and, hence, were excluded from validation of the $Z$-score threshold but were included in the overall calculations of concordance between the BM tumour DNA and cfDNA data across all samples analysed in this study.

In total, 27 coding mutations in BM-derived DNA samples were used to identify true positive and false positive mutation calls in the corresponding 25 cfDNA samples of the training cohort (Fig. 3). We observed significant inter-sample and inter-batch variability in the distribution of the absolute tumour LOD scores (Fig. 4). We addressed this issue by calculating modified $Z$-scores (based on median and median absolute deviation (MAD) in tumour LOD scores for each sample) and used a receiver–operator curve (ROC) to select a modified $Z$-score threshold of 20. This allowed us to set sample-specific tumour LOD score thresholds to filter sequencing artefacts and polymerase errors from real mutation calls and germline polymorphisms (Supplementary Fig. 9). Using this approach, we identified 26 of 27 mutations seen in BM (96% concordance, Supplementary Fig. 10). A *KRAS* p.G12D mutation missed by cfDNA sequencing in MYL-020 had low allele fraction (AF, 1.3%) in the matching BM sample when analysed using 5-gene targeted deep sequencing, which is below the limit of detection of clinical laboratory testing of BM aspirates ($\sim 10\%$). We confirmed the absence of *KRAS* p.G12D mutation in MYL-020 cfDNA by droplet digital PCR (ddPCR), using a commercially available pre-validated PrimePCR ddPCR Mutation Assay (Bio-Rad Canada, Mississauga, Ontario, Canada), as described in Supplementary Methods and Supplementary Fig. 11a. LB-Seq detected one mutation not evident in BM-derived tumour DNA, *PIK3CA* p.Y207\* at 0.28% cfDNA AF in MYL-001. This mutation was not

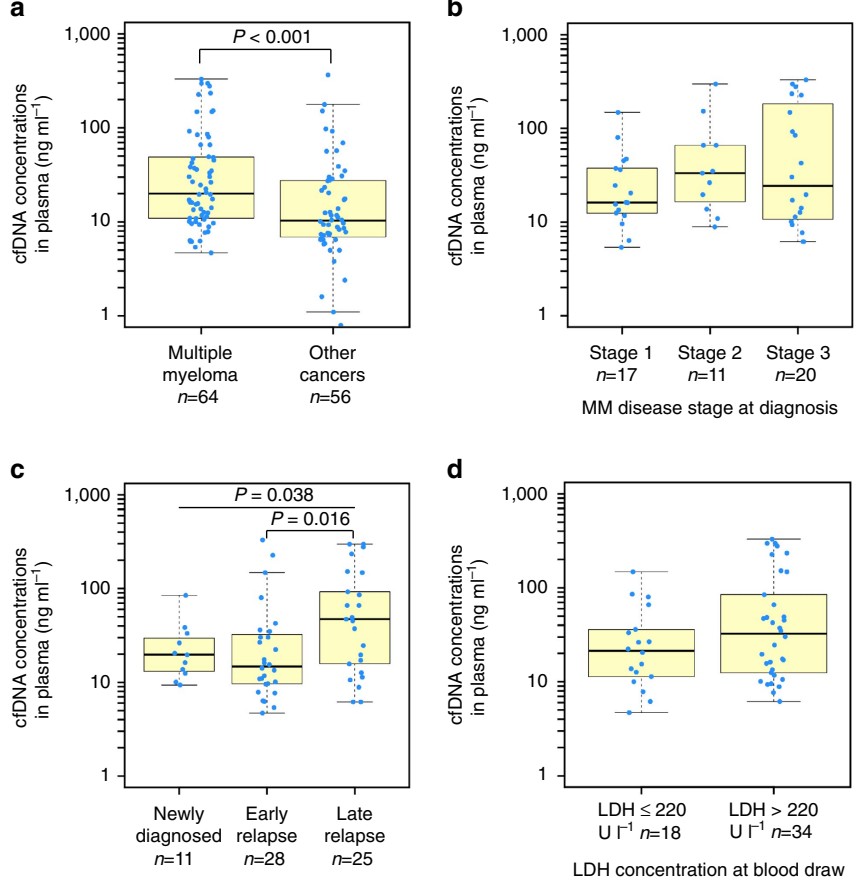

**Figure 1 | Cell-free DNA concentrations in blood plasma of MM patients.** (**a**) Comparison of cfDNA yields in blood plasma samples from patients with MM and other cancer types (Supplementary Table 2). Effect of MM stage at diagnosis (**b**), disease status at blood draw, classified as newly diagnosed or early (1–3 prior lines of therapy) or late relapsed (>3 prior lines of therapy) (**c**), and lactase dehydrogenase concentrations in plasma at blood draw (**d**) on the cfDNA concentrations detectable in blood plasma of MM patients. For each group, the exact sample size is indicated in the figure, underneath group name. All data points represent unique measurements and do not include any technical or biological replicates. The distributions of cfDNA concentrations in each group are shown as box plots, where the central rectangle spans the first to the third quartile (interquartile range or IQR). A segment inside the rectangle shows the median, and 'whiskers' above and below the box show the value 1.5 × IQR above or below the third or the first quartile, respectively. Wilcoxon signed-rank test or Kruskal–Wallis rank sum test was used for comparison of two or multiple groups, respectively, with P value of 0.05 considered statistically significant. Additional clinical measurements and cfDNA yields are provided in Supplementary Data 1.

detected in any other samples except a serial cfDNA sample from the same patient (1.4% AF), while clinical testing of the matching serial BM did not examine this genetic locus.

We next validated our modified Z-score filtering method using a set of 19 matched cfDNA and BM samples from 17 MM patients. We also tested four serial cfDNA samples from patients previously included in the training cohort (marked with *in Fig. 3), which were excluded from validation of the Z-score threshold but were included in the overall calculations of concordance between cfDNA and BM data across all samples analysed in this study. In 19 cfDNA samples included in the validation cohort, LB-Seq detected 19 of 20 likely somatic mutations identified by profiling of matching BM-derived tumour DNA (95% concordance). Similar to the training cohort, the missed mutation (KRAS p.G12V in MYL-054 cfDNA) had low AF (1.3%) in the BM sample (Fig. 3) and we confirmed the absence of this mutation in MYL-054 cfDNA by ddPCR (Supplementary Fig. 11b). LB-Seq also detected a somatic hotspot mutation (PIK3CA p.E545K) in serial cfDNA samples from MYL-049 but not matched BM samples. Allele fractions in the two cfDNA samples (1.0 and 2.1%) were above other verified variants in our cohort (eight other cfDNA mutations with

AF below 1% were concordant with BM data). Furthermore, ddPCR analysis of this mutation (Supplementary Fig. 11c) demonstrated similar AFs in cfDNA and absence of mutation in MYL-049-BM sample, confirming our findings from LB-Seq analysis. One additional mutation, EGFR p.V674F, was also detected in MYL-063 cfDNA but not the matching BM. Compared to training cohort, we observed a trend towards higher mutant allele fractions in samples included in the validation cohort; however, this difference was not significant (P = 0.083 and 0.073 for cfDNA and BM data in Supplementary Fig. 12a,b, respectively). All unfiltered cfDNA sequencing data are available as Supplementary Data 2. Stepwise filtering of candidate mutations is summarized in Supplementary Table 4.

**Batch effect on sequencing data quality**. We identified a strong batch effect on sequencing data quality—the number of variants called in each sample and the absolute LOD scores obtained from muTect analysis. This batch dependence may be related to several factors such as differences in sequencer error rate due to equipment variation or differences in DNA damage during overnight hybridization, sample processing, or storage due to changes in

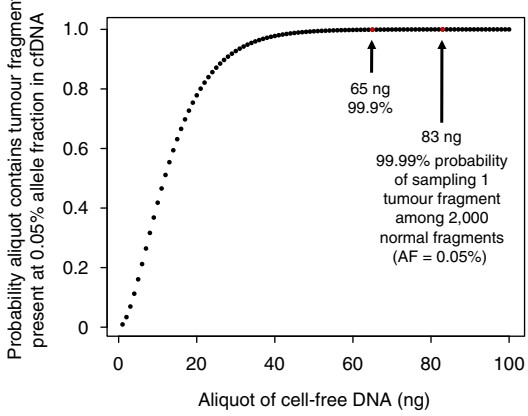

**Figure 2 | Relationship between the amounts of cfDNA used for sequencing and the probability of capturing tumour-derived fragment.** A potential risk for allele drop-out exists when the aliquot of DNA used for these assays is insufficiently large to sample tumour fragments at low concentration within all circulating DNA. The binomial sampling model shown depicts the relationship between the potential risk of allele dropout and the amount of cfDNA used for preparation of DNA sequencing libraries and downstream sequencing. Under a binomial sampling model assuming a haploid genome mass of 3.5 pg, we calculated that to have a 99.99% chance of capturing a tumour fragment at 0.05% concentration, we need to input at least 83 ng of cfDNA into our assay.

reagents, experimental conditions, or personnel[18]. In particular, two batches of samples, for which target capture was done on the same day and that were sequenced in the same sequencing run, had higher absolute LOD scores (median: 23.2; range: 18.5–26.9) compared to previously sequenced training cohort (median: 8.5; range: 6.5–16.5). Coincidentally, 16 of 23 cfDNA samples in the validation cohort were included in the two batches with higher LOD scores (Fig. 4). Despite this potential batch effect, the LB-Seq analysis pipeline was able to effectively differentiate between real genetic variants and sequencing artefacts in each sample by using sample-specific LOD score thresholds. This resilience against batch effect and variation in data quality may be beneficial for using this method in other research laboratories and in clinical setting, but requires additional testing and validation prior to implementation.

**Prevalence of somatic mutations in MM.** Consistent with large-scale genomic studies of MM, BM and cfDNA sequencing revealed wide genetic heterogeneity between patients. Across the training and validation cohorts as well as analysis of 16 cfDNA specimens without matching BM sequencing data available, LB-Seq analysis identified likely somatic mutations in 36 of 53 patients (68%) with mutant AFs ranging from 0.25 to 46% (Fig. 3, Supplementary Fig. 13). We found actionable mutations in 34 of 53 patients (64%) including *KRAS* (38%), *NRAS* (23%) and *BRAF* (11%) variants, some of which co-occurred with *PIK3CA* or *EGFR* mutations. LB-Seq detected several mutations of unknown significance in *EGFR* and *PIK3CA* that were of utility for tracking clonal expansion.

**Comparison of mutant AFs in cfDNA and BM tumour DNA.** Without grouping data by patient, AFs in cfDNA and BM did not correlate ($R^2 = 0.34$, Supplementary Fig. 14), likely due to differences in the relative contribution of tumour DNA to the total cfDNA in plasma of each patient. To test this hypothesis, we examined whether the clonal frequencies of mutations in

BM were recapitulated in circulating tumour DNA in 11 patients with multiple mutations (Fig. 5 and Supplementary Figs 15 and 16). Three of these patients had two serial cfDNA and BM samples each. Overall, mutant AFs were highly concordant between cfDNA and BM (Fig. 5a, $R^2$ range 0.913–0.997). Two patients with mutations in three or more genes had identical subclonal hierarchy determined from cfDNA and BM tumour DNA sequencing (Fig. 5b). Two additional patients had multiple mutations within the same gene with AFs consistent in BM and cfDNA (Fig. 5c). In one such case, we were able to resolve two adjacent base substitutions affecting the same codon leading to distinct *NRAS* p.Q61R and p.Q61K mutations. These two mutations likely originated from different tumour subclones, evident from the lack of overlap between DNA fragments carrying each mutation and marked difference in AFs of these alterations in BM (47% and 1.4%, respectively) and cfDNA (24% and 0.72%, respectively) (Supplementary Fig. 17).

Serial blood draws from three patients (MYL-001, 012 and 049) displayed similar consistency of mutational hierarchy between cfDNA and BM-DNA serial samples. MYL-001 and 049 each had additional *PIK3CA* mutations (p.Y207* and p.E545K, respectively) uncovered by cfDNA analysis (Supplementary Fig. 15) that were not detected in the matching BM aspirates. These mutations were consistent across serial blood draws and not detected in any other cfDNA samples analysed.

**Serial testing of patients treated with trametinib.** Seven patients for which serial sampling was available were enrolled in the PHL-9460 clinical trial of MEK inhibitor trametinib for relapsed/refractory myeloma with at least two prior lines of therapy (NCT01989598). Of these seven patients, two patients (MYL-033 and MYL-049) did not have *KRAS*, *NRAS* or *BRAF* mutations and another two patients (MYL-012 and MYL-022) had samples collected before and at study entry, but no serial cfDNA samples were available after starting trametinib. Only three patients had *KRAS*, *NRAS* or *BRAF* mutations, which we were able to follow in cfDNA and BM-derived tumour DNA during the course of trametinib therapy to evaluate the subclonal response to treatment. Two of these patients progressed with increasing kappa light chains and corresponding increases in AFs of *NRAS* p.Q61K and *KRAS* p.A146T (MYL-003 and MYL-018, respectively). A third patient (MYL-043) progressed on trametinib with increasing kappa light chain concentration and bony and extramedullary disease but decreasing AFs of *KRAS* p.G13D in cfDNA. When AKT inhibitor GSK2141795 was added, we observed discordant response with decrease in kappa light chain concentration and allele fractions of *KRAS* p.G13D in cfDNA and BM, but increase in M-protein (indicative of progression) and growth of a new extramedullary lesion. One of the patients without *KRAS*, *NRAS* or *BRAF* mutations, MYL-049, progressed on trametinib with explosive extramedullary disease. We detected only *PIK3CA* mutations in this patient, including p.H59P mutation (AFs in cfDNA and BM decreased marginally during progression) and subclonal p.E545K mutation only detected in cfDNA (AF increased from 1.0 to 2.1% during progression). The other biomarker-negative patient (MYL-033) had stable disease on study with no mutations detected at screening or follow-up. Clinical characteristics of all other patients are available in Supplementary Data 1 and Supplementary Note 1.

**Discussion**

While BM aspiration is currently required for the diagnosis of MM and for some measures of response to treatment, there are significant limitations particularly in the areas of targeted therapy and minimal residual disease (MRD) monitoring that require

| Cohort | Clinical Lab | CoMMpass | 5-gene panel | Sample ID | KRAS Protein change | KRAS cfDNA allele fraction | KRAS Tumour allele fraction | NRAS Protein change | NRAS cfDNA allele fraction | NRAS Tumour allele fraction | BRAF Protein change | BRAF cfDNA allele fraction | BRAF Tumour allele fraction | EGFR Protein change | EGFR cfDNA allele fraction | EGFR Tumour allele fraction | PIK3CA Protein change | PIK3CA cfDNA allele fraction | PIK3CA Tumour allele fraction |
|---|---|---|---|---|---|---|---|---|---|---|---|---|---|---|---|---|---|---|---|
| Training | ◆ | | ◆ | MYL-018 | p.A146T | 2.0% | 22% | | | | | | | | | | | | |
| | ◆ | | ◆ | MYL-026 | p.Q61H | 2.0% | 29% | | | | | | | | | | | | |
| | ◆ | | ◆ | MYL-030 | p.G13D | 11% | 42% | | | | | | | | | | | | |
| | ◆ | | | MYL-031 | p.K117N | 28% | 49% | | | | | | | | | | | | |
| | ◆ | | | MYL-020 | p.Q61H | 0.53% | 2.8% | | | | | | | | | | | | |
| | | | | | p.G12V | 1.0% | 6.4% | | | | | | | | | | | | |
| | | | | | p.G12D | negative | 1.3% | | | | | | | | | | | | |
| | ◆ | | | MYL-023 | p.Q61H | 1.5% | 5.0% | | | | p.V600E | 1.0% | 2.9% | p.A859S† | 0.41% | 0.8% | | | |
| | | | | | p.G13C | 0.86% | 1.4% | | | | | | | | | | | | |
| | ◆ | | | MYL-012 | p.A146V | 7.2% | 8.0% | | | | p.D594G | 8.5% | 14% | | | | | | |
| | ◆ | | | MYL-007 | p.G13C | 0.50% | 14% | | | | | | | p.C620W | 0.48% | 13% | | | |
| | ◆ | | ◆ | MYL-002 | | | | p.G13D | 0.25% | 0.93% | | | | | | | | | |
| | ◆ | | ◆ | MYL-003 | | | | p.Q61K | 1.3% | 45% | | | | | | | | | |
| | ◆ | | ◆ | MYL-003(2) | | | | p.Q61K | 20% | 61% | | | | | | | | | |
| | ◆ | | ◆ | MYL-019 | | | | p.G13R | 15% | 61% | | | | | | | | | |
| | ◆ | | ◆ | MYL-022 | | | | p.Q61R | 1.2% | 27% | | | | | | | | | |
| | ◆ | | ◆ | MYL-039 | | | | p.Q61R | 1.9% | 34% | | | | | | | | | |
| | ◆ | | ◆ | MYL-028 | | | | p.Q61R | 24% | 47% | | | | | | | | | |
| | ◆ | | ◆ | MYL-001 | | | | p.Q61K | 0.72% | 1.4% | | | | | | | p.I841V | 23% | 26% |
| | | | | | | | | p.Q61K | 32% | 39% | | | | | | | p.Y207* | 0.28% | negative |
| | ◆ | | ◆ | MYL-027 | | | | | | | | | | p.E551K | 41% | 16% | | | |
| | ◆ | | | MYL-004 | | | | | | | | | | | | | | | |
| | | ◆ | | MYL-005 | | | | | | | | | | | | | | | |
| | | ◆ | | MYL-016 | | | | | | | | | | | | | | | |
| | ◆ | | | MYL-033 | | | | | | | | | | | | | | | |
| | | | ◆ | MYL-033(2) | | | | | | | | | | | | | | | |
| | ◆ | | ◆ | MYL-034 | | | | | | | | | | | | | | | |
| | ◆ | | ◆ | MYL-036 | | | | | | | | | | | | | | | |
| | ◆ | | ◆ | MYL-037 | | | | | | | | | | | | | | | |
| Validation | ◆ | | ◆ | MYL-043 | p.G13D | 20% | 8.5% | | | | | | | | | | | | |
| | | | ◆ | MYL-043(3) | p.G13D | 1.4% | 1.1% | | | | | | | | | | | | |
| | | | ◆ | MYL-046 | p.G13R | 6.4% | 42% | | | | | | | | | | | | |
| | | | ◆ | MYL-051 | p.Q61H | 3.4% | 41% | | | | | | | | | | | | |
| | | | ◆ | MYL-054 | p.G12V | negative | 1.3% | | | | | | | | | | | | |
| | | | ◆ | MYL-068 | p.G13D | 46% | 76% | | | | | | | | | | | | |
| | | | ◆ | MYL-058 | p.G12V | 20% | 48% | | | | p.D594N | 15% | 32% | p.E513E | 2.9% | 7.6% | | | |
| | ◆ | | | MYL-012(2)* | p.A146V | 7.6% | 8.0% | | | | p.D594G | 8.2% | 15% | | | | | | |
| | | | ◆ | MYL-070 | p.Q61H | 3.4% | 41% | | | | p.G469V | 0.39% | 3.5% | | | | | | |
| | | | ◆ | MYL-044 | p.G13D | 17% | 41% | | | | | | | p.C624Y | 2.9% | 5.7% | | | |
| | | | ◆ | MYL-063 | p.Q22K | 4.6% | 38% | | | | | | | p.V674F | 0.64% | negative | | | |
| | | | ◆ | MYL-055 | | | | p.Q61K | 4.4% | 40% | | | | | | | | | |
| | | | ◆ | MYL-022(2)* | | | | p.Q61R | 3.0% | 42% | | | | | | | | | |
| | | | ◆ | MYL-084 | | | | p.G13R | 0.73% | 46% | | | | | | | | | |
| | ◆ | | | MYL-001(2)* | | | | p.Q61K | 26% | 47% | | | | | | | p.I841V | 22% | not tested |
| | | | | | | | | | | | | | | | | | p.Y207* | 1.4% | not tested |
| | | | ◆ | MYL-056 | | | | | | | p.D594N | 2.6% | 45% | | | | | | |
| | | | | | | | | | | | p.A322T | 45% | 50% | | | | | | |
| | | | ◆ | MYL-027(2)* | | | | | | | | | | p.E551K | 46% | 52% | | | |
| | | | ◆ | MYL-049 | | | | | | | | | | | | | p.H59P | 40% | 41% |
| | | | | | | | | | | | | | | | | | p.E545K | 1.0% | negative |
| | | | ◆ | MYL-049(2) | | | | | | | | | | | | | p.H59P | 38% | 36% |
| | | | | | | | | | | | | | | | | | p.E545K | 2.1% | negative |
| | ◆ | | | MYL-017 | | | | | | | | | | p.A1201T | 0.31% | not tested | | | |
| | | | ◆ | MYL-050 | | | | | | | | | | | | | | | |
| | | | ◆ | MYL-053 | | | | | | | | | | | | | | | |
| | ◆ | | | MYL-077 | | | | | | | | | | | | | | | |

Legend:
- Cyan: Concordant between cfDNA and BM (true positive)
- White: No mutations detected in cfDNA or BM (true negative)
- Yellow: Detected in cfDNA but not BM
- Orange: Detected only in BM (false negative)
- Grey: Only cfDNA data available, mutations not profiled in BM

**Figure 3 | Comparison of mutations identified by cfDNA sequence analysis and genetic profiling of BM-derived tumour DNA.** Sample IDs followed by (2) or (3) represent serial samples obtained on a separate clinical visit. EGFR protein changes are annotated using ENST00000275493 transcript. *Serial plasma samples from patients included in the training cohort that were excluded from validation of the Z-score threshold but were included in the overall calculations of sensitivity (concordance between LB-Seq and BM tumour DNA profiling data) and specificity of mutation calling across all samples analysed in this study (Supplementary Fig. 10); †Single amino acid substitution resulting from a two-base genomic substitution. For BM samples with data from more than one source (specified by black diamond in BM analysis column), all sequencing results are presented in Supplementary Table 1.

repeated BM sampling. The requirement for BM aspiration is limited not only by patient discomfort but technical difficulties in sampling and variability of myeloma distribution within the marrow, at times resulting in suboptimal aspirate samples precluding tumour genetic profiling. We present here validation of LB-Seq, a circulating tumour DNA sequencing assay to screen for mutations throughout a diversity of genomic regions (for example, exons) rather than mutation hotspots, that is comparable to current clinical profiling of tumour specimens and may become a valuable adjunct to BM testing in MM.

LB-Seq mutation calls correctly predicted 96% of mutations detected in matching BM-derived tumour DNA samples with >98% specificity. The absence of two mutations apparently missed by LB-Seq in cfDNA was confirmed by ddPCR analysis. Furthermore, these mutations were supported by only 1.3% of reads in the corresponding BM-derived tumour DNA samples

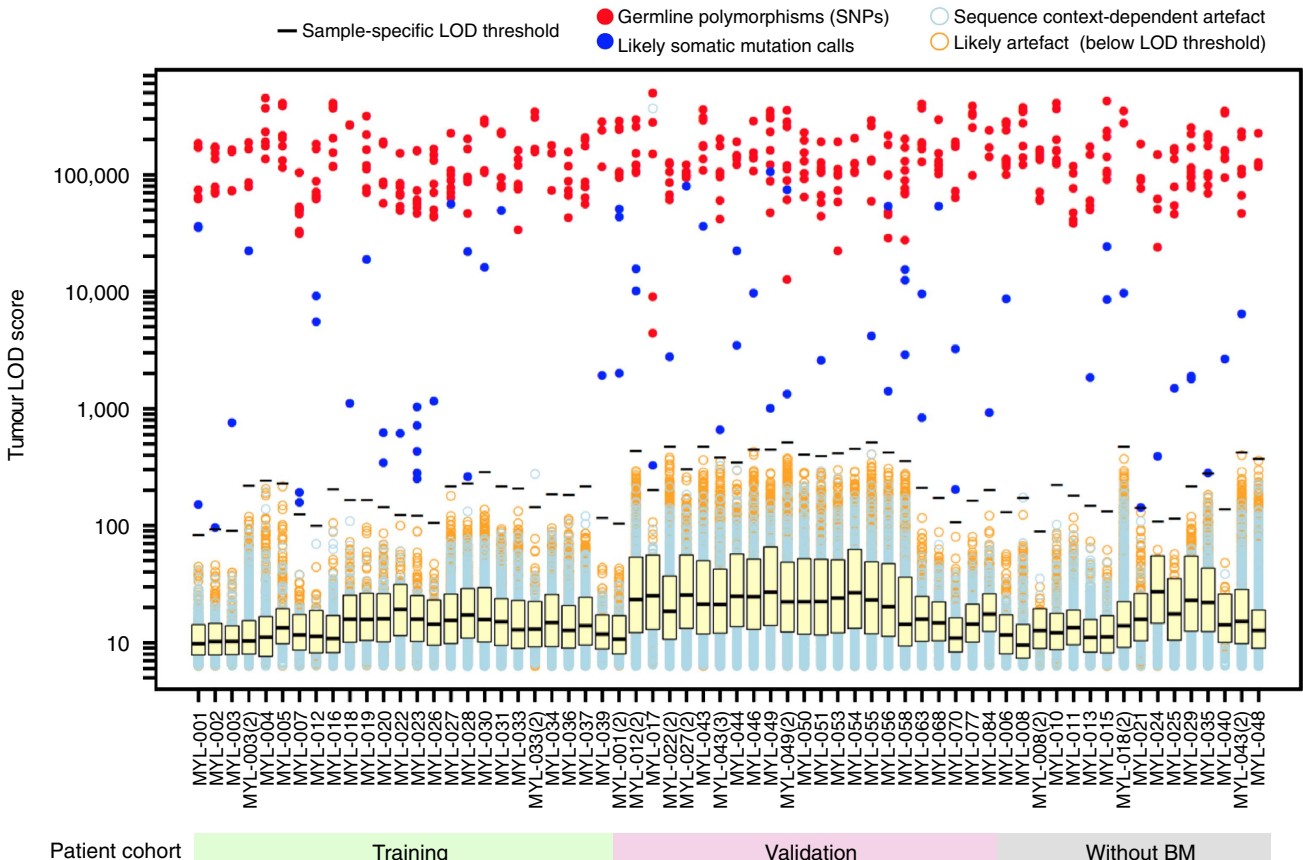

**Figure 4 | Distribution of tumour LOD scores in cfDNA sequencing data.** For each sample, all candidate mutation calls generated by muTect version 1.1.4 were divided into subgroups based on the type of mutation or the filtering step at which they were removed, as indicated in the legend. The stripcharts showing the tumour LOD scores for each subgroup of mutations were overlaid with the boxplot demonstrating the distribution of tumour LOD scores for all mutation calls kept by muTect within each sample, prior to downstream filtering. The central rectangle spans the first to the third quartile (IQR), and segment inside the rectangle shows the median tumour LOD score for each sample. Sample-specific thresholds for calling likely somatic mutations ($-$) were determined as the tumour LOD score corresponding to the modified $Z$-score of 20 (that is, 20 MADs above the median). Unfiltered annotated data for all samples are available as Supplementary Data 2. Custom Rscripts (in R version 3.2.2) for filtering and plotting LOD score distribution data are available at www.github.com/pughlab/lb-seq.

measured by ultra-deep sequencing and were not reported by the clinical laboratory due to low allele fraction. Hence, LB-Seq detected all tumour-derived mutations detected by the clinical laboratory testing of BM aspirates.

The prevalence of *KRAS*, *NRAS* and *BRAF* mutations in our patient cohort (38%, 23% and 11%, respectively) was higher compared to previously published data (23–26%, 20–24% and 4–6% of MM cases, respectively) obtained from shallow genome (30 $\times$) and whole exome (100 $\times$) sequencing studies in MM patients[3,4]. These differences may be explained by increased sensitivity of our method, since 37 of 48 BM tumour DNA samples analysed in this study were sequenced using the 5-gene targeted sequencing approach (> 5,000 $\times$ mean target coverage). As a result, we were able to detect many mutations with AFs < 10%, including 10 of 22 *KRAS* mutations, 2 of 13 *NRAS* mutations and 2 of 7 *BRAF* mutations that most likely would be missed by shallow genome/whole exome sequencing. In addition, treated MM patients are reported to have significantly higher AFs for somatic mutations in recurrently mutated genes such as *KRAS*, *NRAS* and *BRAF* when compared to untreated patients[4]. This may lead to improved rate of detection of somatic variants in these genes in treated MM patients, who make up 83% of our study cohort but only 50% of the cohort in studies reporting lower prevalence of somatic mutations in these genes[3,4].

We were able to perform genetic profiling of seven plasma samples for which BM aspiration yielded insufficient malignant plasma cells for conventional sequencing methods, as well as 11 plasma samples for which BM aspirates were not collected. Furthermore, all 13 patients enrolled on the clinical trial of trametinib were correctly identified as biomarker positive or negative using cfDNA analysis. This demonstrates the potential for a minimally invasive platform with broader applicability for identifying patients that may derive benefit from inclusion in targeted therapy trials and the possibility of simplifying trial recruitment or stratification using LB-Seq.

We analysed serial cfDNA samples from patients enrolled in trametinib trial during the course of therapy to investigate the role of MAPK pathway mutations in patient response to trametinib treatment. In three patients with evidence of *KRAS*, *NRAS* or *BRAF* mutations and serial plasma samples available during the course of trametinib treatment, we observed a good correlation between data from sequencing of cfDNA and matching BM-derived tumour DNA but some inconstancies between clinical and molecular response to treatment. In two patients, clinical disease progression was associated with an increase in AFs of *NRAS* and *KRAS* mutations, in agreement with our prediction of an MAPK pathway-dependent mechanism of resistance to trametinib. However, the third patient demonstrated

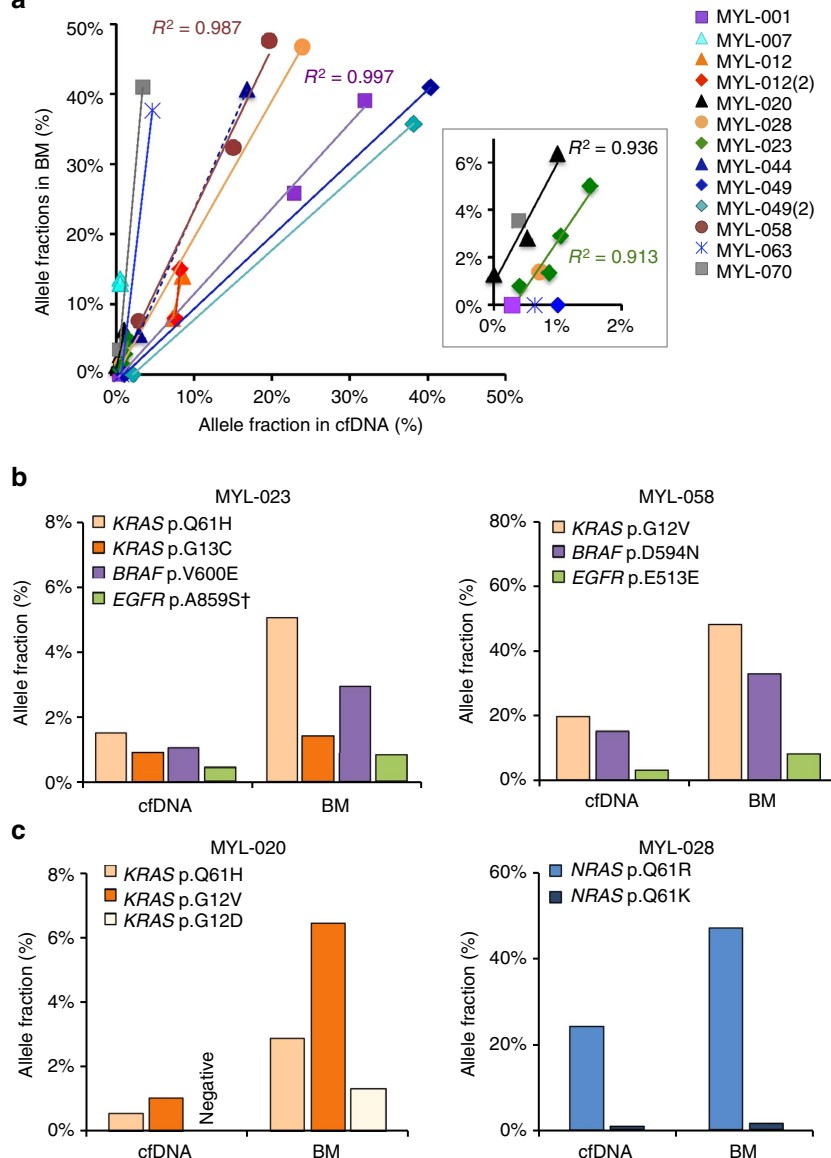

**Figure 5 | Comparison of mutant allele frequencies in cfDNA and BM-derived tumour DNA.** For 11 subjects with multiple mutations in one or more genes, a scatter plot of cfDNA and BM-derived DNA allele fractions was generated for each mutation in each subject. Linear regression was then assessed for patients with three or more mutations to determine the strength of correlation ($R^2$ value) (**a**). Comparison of relative tumour and cfDNA AFs and clonal hierarchies determined from cfDNA and BM in two patients, MYL-023 and 058, with three or more mutations in multiple genes (**b**) and two patients, MYL-020 and 028, with multiple mutations in the same gene (**c**). AF correlation plot for all cfDNA samples with matching BM samples ($n = 48$) and bar graphs for other samples with multiple mutations excluded from **b**,**c** of this figure are presented as Supplementary Figures 14–16.

clinical progression, with increase in M-protein and growth of a new extramedullary lesion, but a decrease in AFs of *KRAS* p.G13D mutation in BM and cfDNA. Relatively low AF in the BM tumour DNA (19%) suggests that *KRAS* p.G13D mutation in this patient is likely subclonal. Hence, we suspect that the observed clinical progression may be caused by a differential response to treatment within the subclones, with effective trametinib response in *KRAS*-mutated subclone but not the non-mutated subclones, or appearance of new secondary resistance mutations in genes we did not examine using our 5-gene panel analysis. To confirm our hypothesis, this question needs to be further investigated using a more comprehensive genetic profiling of MM tumour DNA and cfDNA in serial samples from a larger patient cohort. Nonetheless, our

current data indicate that cfDNA sequencing may serve as a surrogate for serial sampling of the BM for monitoring patients on trial.

Molecular profiling of CD138-selected malignant plasma cells from single site BM aspirates may inadequately reflect the clonal heterogeneity and patchy distribution of MM. There is some evidence that cfDNA samples may provide insights into subclonal architecture in other tumours[19], and our study supports this finding in MM, a disease characterized by a complex subclonal structure[4,20–22]. Strikingly, the relative subclonal composition of MM cells in BM appears to be reflected in mutant AFs in cfDNA obtained using LB-Seq from cases with multiple mutations in single or multiple genes. These findings suggest cfDNA sequence analysis has the potential to enable reconstruction of subclonal

hierarchies using relative AFs from blood plasma and further investigation is warranted.

In two patients, LB-Seq identified *PIK3CA* mutations not found by BM sequencing but persistently found in serial blood draws at AFs well above many other true mutations detected by LB-Seq in other samples. One of these mutations, *PIK3CA* p.E545K, is a recurrent somatic mutation detected across many cancer types. This mutation was not detectible in any other 62 cfDNA samples sequenced in this study, and we confirmed the presence of this mutation in both cfDNA samples from MYL-049 but not the matched BM sample by ddPCR analysis. These results suggest that this is a real mutation present in MYL-049 cfDNA but absent in the BM samples used for comparison. Clinical correlation revealed that this patient had progressed to extramedullary disease with retroperitoneal adenopathy, splenomegaly and multiple sites within the pancreas, adrenals and liver. Extramedullary tumours were unavailable for sequencing; however, the clinical picture of disease progression is compelling evidence for cfDNA arising from one of (i) extramedullary disease; (ii) progression in BM not sampled by BM aspiration; or (iii) tumour subclones not adequately represented in the BM aspirate. Similarly, a *PIK3CA* p.Y207* mutation detected only in serial cfDNA samples from MYL-001 likely represents a real tumour-derived mutation not adequately represented in the localized BM aspirate used for comparison or originating from the metastatic tumour sites. The third variant identified only in cfDNA of MYL-063 patient (*EGFR* p.V674F) has been previously reported in lung cancer[23] and may represent a true mutation missed by BM sequencing; however, we did not sequence serial cfDNA samples from this patient to confirm the persistence of this mutation. While there may be similar limitations to cfDNA sampling that have not yet been clearly defined, these cases highlight the potential for cfDNA testing to capture mutations in clones not represented by single BM aspirates and to complement BM sampling in studies of subclonality and MRD assessment moving forward.

Improved sensitivity and the potential for subclonal structure monitoring make the LB-Seq approach ideal for serial monitoring of MM, minimizing the need for medically unnecessary BM sampling. Although we identified several potentially somatic mutations in *EGFR* and *PIK3CA* of unknown clinical significance in MM, these mutations may be markers of clonal evolution of this disease or response to therapy. These findings suggest that LB-Seq may allow earlier detection of secondary resistance mutations and monitoring of dynamic changes in AFs of clonal and subclonal mutations under various therapeutic pressures in a typical course of therapy. These potential applications would require a more comprehensive longitudinal analysis of LB-Seq in a larger set of serial cfDNA samples from relapsed or newly diagnosed MM patients with matching BM sequencing data. We anticipate that the sensitivity of LB-Seq can be further increased by applying molecular barcoding methods to help distinguish true somatic mutations with extremely low AFs from sequencing artefacts and polymerase errors[24,25]. In addition, LB-Seq method can be extended to larger target capture panels that include many recurrently mutated genes that correlate with disease prognosis in MM and genes associated with response or secondary resistance mutations to MM therapies. Together, the use of molecular barcoding and expanded target region for LB-Seq may allow development of a sensitive and non-invasive test for detection of MRD in MM.

In this study, we observed higher cfDNA extraction yields in MM patients compared to those with advanced solid tumours, potentially due to higher shedding of tumour DNA into circulation from malignant plasma cells. Increased levels of circulating tumour DNA in advanced stage MM may also explain

relatively high AFs of tumour-derived mutations (11% average, 0.25–46% range) measured by LB-Seq in this study population. Although comparable cfDNA yields and mutant AFs have been reported in other advanced tumours[12,15,26], some cancer types may have lower yields of cfDNA and lower AFs for tumour-specific mutations detectable by LB-Seq. This may affect the feasibility of utilizing LB-Seq to achieve the comparable sensitivity in other patient populations or necessitate collection of increased volumes of plasma.

Our method differs from previous protocols in several important aspects. Rather than using a common approach of looking only at mutation hotspots, we targeted all protein-coding exons of genes of interest to allow in a single assay, simultaneous detection and tracking of any mutations in the target region over the course of therapy[27,28]. This is of importance in MM, a disease characterized by an abundance of subclonal genetic abnormalities (including adjacent mutations affecting the same codon within distinct subclones), high degree of intra- and inter-patient genetic heterogeneity and treatment-associated secondary resistance mutations[1–4]. We also implemented and validated an analytic approach for analysis of cfDNA sequencing data that allowed detection of tumour-derived mutations in cfDNA with AFs as low as 0.25% with >98% specificity and 96% concordance with mutations detected by the state-of-the-art molecular profiling of the matching BM tumours.

Unlike amplicon-based sequencing, our hybridization-based method retains the fragment size distribution seen in native cfDNA, potentially enabling fine dissection of fragmentation sites and nucleosome positioning within DNA circulating in blood[29]. A non-invasive epigenetic profile for cancer is an attractive future direction for capture-based cfDNA diagnostics, particularly for diseases with highly specific epigenetic alterations or not necessarily driven by somatic mutation alone.

While we included only five genes in this study, we expect that LB-Seq is scalable for examination of larger portions of the genome to query other target genes or mutation classes (such as rearrangements and copy number alterations), as proof-of-concept studies of cfDNA exome and genome sequencing have been reported in other disease sites[26,28]. With increasing panel size, the sensitivity and specificity of this approach may be affected by poor mapping quality in specific genomic regions (for example, pseudogenes, high homology regions), poor capture efficiency or coverage uniformity, increased sequence bias, and increased cost of sequencing necessary to achieve a similar depth of coverage. Some of these challenges may be overcome by parallel sequencing of matched normal DNA and/or cfDNA from healthy volunteers as well as further improvement of library preparation and target capture methodology to allow identification and removal of systematic mapping artefacts and polymerase or sequencer errors (for example, applying duplex sequencing method with molecular barcoding and error suppression strategies)[16,24,25]. Furthermore, to optimize the sensitivity and specificity of LB-Seq with a new target capture panel design, the appropriate modified Z-score threshold for differentiating between real mutation calls and sequencing artefacts should be determined empirically through the use of the ROC curve and training and validation cohorts, as described in this study.

With increasing understanding of MM genetic landscape characterized by recurrent IgH translocations, copy number aberrations and somatic point mutations with proven prognostic significance or links to treatment response, this approach can provide an unprecedented opportunity for comprehensive profiling of MM tumour, real-time monitoring of patients receiving therapy, and earlier detection of disease progression or MRD using a non-invasive blood-based assay[1,2,4,30,31]. Hence,

this approach represents a significant advancement for molecular profiling of patients with MM and cancers in general.

## Methods

**Study design.** This study was designed in parallel to an ongoing Phase II clinical trial of the MEK inhibitor trametinib, in which eligible participants are stratified into biomarker-positive or negative groups based on clinical molecular genetic testing for activating mutations within *KRAS*, *NRAS* or *BRAF* (PHL-9460, NCT01989598, protocol approved by the Ontario Cancer Research Ethics Board). Two additional genes, *EGFR* and *PIK3CA*, were included in the analysis as part of an existing 5-gene exon-capture panel developed by our laboratory. To increase the power of cfDNA analysis, we enrolled an additional 40 patients with active myeloma undergoing routine clinical care. Informed consent from all study participants was obtained before blood and BM collection in accordance with the University Health Network Institutional Review Board approved study protocol (Certificate 14-774-CE).

**Collection of blood and bone marrow specimens.** Ten subjects provided two ($n = 9$) or three ($n = 1$) serial blood samples obtained during separate clinic visits (range: 1.1–8.6 months apart), while 43 other participants provided a single blood sample for a total of 64 samples. In this set, 48 cfDNA samples from 39 patients had BM-derived tumour DNA sequencing data available from BM aspirates drawn for enrolment onto the PHL-9460 clinical trial or as standard-of-care. Patient demographics and disease characteristics are presented in Table 1 and Supplementary Data 1.

Tumour DNA was obtained from CD138 + cells isolated from BM aspirates, as described previously[32]. BM-derived tumour DNA was sequenced at the CAP/CLIA-certified clinical laboratory at the Princess Margaret Cancer Centre ($n = 31$) or as part of the myeloma molecular profiling CoMMpass study ($n = 5$, www.themmrf.org/research-partners/the-commpass-study). When possible, DNA from BM specimens were sequenced or re-sequenced in our laboratory using 5-gene target capture panel ($n = 37$) to allow direct comparison with cfDNA sequencing data (Supplementary Table 1).

**LB-Seq laboratory workflow.** Circulating cfDNA was extracted from blood plasma collected within 1 h of blood draw using the QIAamp Circulating Nucleic Acid kit (Qiagen, Valencia, CA, USA) (Supplementary Methods). To prepare cfDNA-sequencing (cfDNA-seq) libraries, we used 83 ng of input cfDNA (based on mathematical estimate in Fig. 2) or maximum amount available (10–80 ng, 13 samples). These libraries were prepared using KAPA Hyper Prep Kit for Illumina TruSeq library construction (Kapa Biosystems, Wilmington, MA, USA) in conjunction with one of two types of single-indexing adapters, Illumina TruSeq LT adapters (Illumina, San Diego, CA, USA) or NEXTflex-96 DNA Barcodes (Bio Scientific, Austin, TX, USA) (Supplementary Figs 1–4).

To enrich cfDNA-seq libraries for genomic regions of interest, we designed 146 synthetic, 120 nucleotide-long single-stranded DNA biotinylated capture probes targeting all protein-coding exons of *KRAS*, *NRAS*, *BRAF*, *EGFR* and *PIK3CA* with $1 \times$ tiling density (xGen Lockdown Custom Probes Mini Pool, Integrated DNA Technologies, Coralville, IA, USA). Prior to hybrid capture, 8–11 barcoded cfDNA-seq libraries were pooled in equal concentrations for a total of 500–1,000 ng of DNA. We followed manufacturer protocols for probe hybridization, target capture, post-capture amplification and bead clean up of captured amplified DNA, with minor modifications such as increased hybridization time (Supplementary Methods). We generated paired-end 100 base pair sequencing reads using Illumina HiSeq 2000 or HiSeq 2500 instruments, pooling 20–25 samples from multiple hybridization batches per flow cell lane to achieve mean bait coverage $> 20,000 \times$ (Supplementary Fig. 5).

**LB-Seq bioinformatics pipeline and data analysis.** Unique sample barcodes (6 or 8 nucleotide indexes) were used to de-multiplex raw sequencing data and generate sample-specific FASTQ files. Reads were aligned to the human genome reference (Illumina iGenomes hg19) using bwa (ref. 33) version 0.7.12 and post-processed following Genome Analysis Toolkit (GATK) Best Practices[34] without marked duplicates, to avoid mistakenly reducing the true library complexity, as described in Supplementary Methods and CAPP-Seq protocol[14]. Mean bait coverage was estimated using GATK Depth of Coverage analysis, counting the number of fragments rather than bases. We used Picard[35] CalculateHsMetrics tool to assess data quality (total reads, on-bait, near-bait and off-bait reads, percentage of selected bases) and CollectInsertSizeMetrics tool to evaluate the inferred insert size of the DNA sequencing libraries (Supplementary Fig. 4c).

Candidate point mutations were called using muTect version 1.1.4 (ref. 36) configured to allow detection of rare variants ($< 1\%$) in ultra-deep sequencing data without a matching normal (Supplementary Figs 6 and 7). In this study, we configured muTect for analysis of low frequency mutations in cfDNA by setting 'fraction_contamination' to 0 to enable emission of low frequency variants, setting 'gap_events_threshold' to 1,000 to increase the number of reads tolerated with indels in high coverage data, and enabling the 'force_alleles' flag to prevent rejection of mutations due to triallelic sites (see details in Supplementary Methods

section). To distinguish real genetic variants (*de novo* somatic mutations and germline polymorphisms) from sequencing artefacts or polymerase errors, we developed a data-filtering algorithm in R version 3.2.2 (available at www.github.com/pughlab/lb-seq). This filtering algorithm relies on the use of tumour LOD scores, defined as the Log of (likelihood tumour event is real/ likelihood event is sequencing error), to differentiate true mutations present at very low allele frequencies from sequencing artefacts with similar apparent allele frequencies[36]. To account for variability of the absolute tumour LOD score values between samples and sequencing batches, tumour LOD scores were transformed into modified *Z*-scores, that is, the distance from the median divided by the MAD[37] for each sample, which are independent of the normality assumption. Similar approach has been described previously for transformation of high throughput screening data generated for large-scale RNA interference libraries[38]. Since the use of this approach for transforming tumour LOD score distribution data has not been described previously, we used the ROC generated using the training cohort of 25 cfDNA samples with matching BM data available to establish the threshold value for the modified *Z*-score, which would allow us to differentiate true mutations from sequencing artefacts (Supplementary Fig. 9). From ROC analysis, the modified *Z*-score of 20 (that is, tumour LOD score 20 MADs above the median LOD score in each sample) achieved optimal sensitivity and specificity for detection of mutations identified in the matching BM samples and was used to identify *bona fide* variants in all other samples.

We next removed likely germline polymorphisms present in $> 0.1\%$ of populations curated by dbSNP (build 142) or the 1000 Genomes Project (1000gp3 20130502 database), as annotated by Oncotator version 1.5.3.0 (ref. 39). While this approach allowed us to differentiate between germline SNPs and tumour-derived somatic variants without matching normal DNA, it is possible that rare germline variants not represented in the publically available population variation databases may not be appropriately filtered by our bioinformatics pipeline. To ensure accurate identification of germline variants, we recommend sequencing matched normal genomic DNA obtained from leukocytes readily available from the same blood samples used for cfDNA extraction, especially in patients with evidence of high-AF mutation calls ($> 40\%$) or when applying this method to a larger target panel design. Unfiltered mutation calls kept by muTect and annotated by Oncotator for all 64 cfDNA samples are included as Supplementary Data 2.

**Droplet digital PCR analysis.** To confirm discordant mutation calls obtained from LB-Seq analysis, we used commercially available PrimePCR ddPCR (droplet digital PCR) Mutation Assays and ddPCR reagents (Bio-Rad Canada, Mississauga, Ontario, Canada) to detect each mutation of interest in 25–50 ng of extracted DNA or the corresponding adapter-ligated DNA library, according to the manufacturer's protocol. We used pre-validated *KRAS* p.G12D, *KRAS* p.G12V or *PIK3CA* p.E545K assay for mutation detection in MYL-020 cfDNA, MYL-054 cfDNA or MYL-049 BM, respectively. HEX-labelled probe for WT allele and FAM-labelled probe for mutant allele were combined with sample input DNA prior to generating droplets using QX200 Droplet Generator (Bio-Rad). After PCR amplification of DNA in each droplet using C1000Touch Thermal Cycler (Bio-Rad), the fluorescence signal for each probe was simultaneously measured by QX200 Droplet Reader (Bio-Rad) to assess mutant allele fractions using QuantaSoft version 1.7.4, according to the manufacturer's instructions. A different positive control was used for each assay: MYL-015 cfDNA for *KRAS* p.G12D; MYL-058 cfDNA for *KRAS* p.G12V; and two cfDNA samples from MYL-049 for *PIK3CA* p.E545K. In all three assays, MYL-021-cfDNA sample was used as a negative control and water was used in the no-template control reactions.

**Statistics.** We used Wilcoxon signed-rank test or Kruskal–Wallis rank sum test for comparison of two or multiple groups, respectively, with *P* value of 0.05 considered statistically significant.

**Data availability.** All data that support the findings of this study are available within the paper and its supplementary information files.

All codes used for stepwise filtering of cfDNA sequencing data and plotting the LOD score distribution data in Fig. 4 are available at www.github.com/pughlab/lb-seq and are described in detail in Supplementary Methods section and Supplementary Figs 6 and 7.

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

## Acknowledgements

We thank all study participants, including those enrolled in the PHL-9460 clinical trial designed by S.T. and funded by an NO1 NIH grant held by A.M.O., for providing plasma and BM samples for analysis in this study. We also thank the staff of the Princess Margaret Genomics Centre (Neil Winegarden, Julissa Tsao and Nick Khuu) and Bioinformatics Services (Carl Virtanen, Zhibin Lu and Natalie Stickle) for their expertise in generating the sequencing data used in this study. Thanks also to Dr. Wei Xu, Principal Biostatistician and Scientist at the Princess Margaret Cancer Centre, for his advice on statistical data analysis and interpretation. This work was funded by the Princess Margaret Cancer Foundation (T.J.P., S.T.), Canada Foundation for Innovation, Leaders Opportunity Fund, CFI #32383 (T.J.P.); Ontario Ministry of Research and Innovation, Ontario Research Fund Small Infrastructure Program (T.J.P.); Canadian Cancer Society Research Institute (S.T.); Myeloma Canada (S.T.); and a Multiple Myeloma Research Foundation Research Fellow Award (O.K.).

## Author contributions

O.K. prepared cfDNA and BM-derived DNA samples for sequencing, analysed sequencing data, developed and validated cfDNA data analysis algorithm, and prepared manuscript tables and figures. R.K. prepared study protocol, obtained ethics approval, enrolled patients into the study, collected and summarized clinical and demographic patient data, collected blood samples, extracted blood plasma, and collected BM aspirate samples. A.D. provided computational support in development and validation of cfDNA data analysis algorithm and preparation of figures. T.L., M.D., J.L. and M.M. contributed to preparation of cfDNA samples for sequencing. Z.L. performed CD138 + selection of BM plasma cells and extraction of BM-derived DNA. E.M.-K. contributed to collection of clinical and demographic patient characteristics. T.Z. and S.K.-R. performed sequencing of BM-derived tumour DNA in the clinical laboratory. S.T. designed the study and provided oversight of clinical aspects of the study and protocol development and implementation. T.J.P. provided oversight of laboratory aspects of the study including development and application of methods pertaining to cfDNA extraction, capture, sequencing, and computational approaches for data analysis and interpretation. O.K. drafted the manuscript, with significant input from R.K., S.C., T.J.P. and S.T. All authors reviewed the manuscript prior to submission.

## Additional information

**Competing interests:** S.T. received research support from GlaxoSmithKline, Amgen, Oncoethix and Astellas and provided consultant services to Novartis. T.J.P. received research support from Boehringer Ingelheim. The remaining authors declare no competing financial interests.

