## [Peer Review File · Nature Communications]

Reviewers' comments:

Reviewer #2 (Remarks to the Author):

The manuscript from Kis et al is well timed, well written and with some revision will make an important impact in the burgeoning field of liquid biopsy. The rationale for cfDNA analysis in the management of multiple myeloma (MM) with personalised treatment approaches is clear, where replacing bone marrow aspiration will be of significant benefit to patients, who would then not have to suffer what is often a painful procedure that is not always without risk. The authors set out to make this case by assessing cfDNA in 48 MM patients sequencing all protein coding regions of 5 oncogenes relevant to this disease. The approach is not novel of course, but the application to MM is important and there are no published high quality studies of cfDNA in MM to my knowledge. Importantly, the authors compare cfDNA to matched bone marrow aspirates and report concordance of 44/46 mutations detected. A significant aspect of the manuscript is the development of the bioinformatics pipeline to deal with minimising false data. The case studies they present are also very interesting and well described.

However, the authors make some overstated claims about their data which need to be revisited.

1. They have not incorporated germline analysis that most in the field of cfDNA would consider mandatory. Without this built into their workflow, however good their informatics may be, it is not possible to say with complete surety that they pick up somatic mutations. They should either provide and incorporate germline data into their analysis or refer throughout their manuscript to potential somatic mutations.

2. The authors refer to the inter-patient variability in ctDNA yields that they observe as unusual but this is just as likely to reflect a relatively low sample number and others have reported a similar level of variability which the authors should refer to.
e.g., Establishment of tumor-specific copy number alterations from plasma DNA of patients with cancer. Heitzer et al Int J Cancer. 2013.

3. The allele frequency (AFs) the authors report are also not "unusually" high. e.g. Noninvasive Identification and Monitoring of Cancer Mutations by Targeted Deep Sequencing of Plasma DNA. Forsheo et al. Science Translational Medicine 2012.

4. The method the authors use for targeted sequencing is not really that novel and other publications have also sequenced all protein coding exons of genes, e.g., Tumor-associated copy number changes in the circulation of patients with prostate cancer identified through whole-genome sequencing. Heitzer et al Genome Medicine 2013 and High-Throughput Detection of Actionable Genomic Alterations in Clinical Tumor Samples by Targeted, Massively Parallel Sequencing, Wagle et al Cancer Discovery 2012.

Finally the authors should comment further on the scalability of their approach, adding detail on the maintenance of sensitivity as the gene panel under study expands.

With the text revisions above resolved, this manuscript is worthy of publication in Nat Comms in my opinion.

Reviewer #3 (Remarks to the Author):

The manuscript "Circulating tumor DNA sequences as an alternative to multiple myeloma bone marrow aspirates" by Kris et al. compared mutation profiles detected in bone marrow to those that can be established from circulating cell free DNA. After setting up the methodology by using a training and validation set, the authors find that cfDNA analysis in most cases accurately reflects

the mutational status detected by analysis of the bone marrow. Overall the authors conclude that cfDNA analysis provides a reasonable alternative to bone marrow aspirates.

The manuscript is very well written and clearly shows what a remarkable effort has gone into this work. There are a number of minor and major points that should be addressed before publication.

Major Points:

1) Clarity of the methodology. The authors do an outstanding job in being transparent about their work. In the supplemental information they provide data as well as analytical scripts so their work can be replicated if so desired. The challenge to date is that given the current limitations on review time and resources a comprehensive validation of these techniques is practically impossible. Ultra sensitive detection of DNA mutations from cfDNA by using next generation sequencing is not a widely adapted method today and still in its infancy. Many analytical errors that can be made are hard to uncover, but have severe consequences on the results. Since it is unlikely that we will see a shift in the review process for biological papers, that more closely resembles that review of mathematical papers anytime soon, I recommend that the authors provide a bit more detail in the main text about the analytical methods, alternatively provide a more detailed write-up about the rationale behind their analytical steps. In its current form it is hard to judge the quality of the analytical pipeline. On the surface some items are cause for concern:

a. Use of MuTect as primary variant identification. The chosen setting for MuTect appear to be correct, but MuTect traditionally has underperformed in settings where sub 5% mutations need to be detected (compared to specialized solutions)

b. Please clarify the first step in the R filtering described in the supplement: "Remove mutation calls rejected by MuTect or supported by less than 10% of total reads at that locus". It currently reads as if mutations are rejected that have less than 10% of all reads at this position showing the mutation, which in turn makes it hard to understand how mutations in the sub 10% are detected without the use of single molecular barcodes. Maybe this is a simple typo and it should read "10 total reads"

2) Training and Validation. The authors choose a common model including training, test and validation sets.

a. Supplement Figure 8 shows the use of 59 specimen analyzed in these three phases. The 59 specimen are derived from 48 samples. A total of 10 samples contributed more than one specimen. These specimens are no longer independent and hence their information to some level redundant (from figure 4 it appears at least 4 to 6 samples are affected in the validation set). For example, if all 10 samples, with more than one specimen available, are part of the 16 specimen used in the "without BM" set, the set would no longer qualify as truly independent, because a major subset of specimen had been used to train on. This point should be clarified, ideally the analysis should be repeated without the use of multiple aliquots from one sample. Even though the numbers will be smaller, it is likely that the results will be similar.

b. Batch effect for Tumor LOD scores. Figure 4 shows the LOD scores for all 59 specimens. There is a clear batch effect displayed between the three groups. The baseline in the training model appears to be lower compared to the testing populations. It would be necessary to explain the reasons for the observed effect and why it is not a cause for concern.

3) Analysis.

a. The use of 20 as a cutoff for the modified Z-scores appears to be derived from empirical data. There is a certain risk for overfitting (see above), consequently it would be good to explain why a Z-score of 20 is justifiable from a theoretical basis or why the standard model does not apply. To achieve specificity High specificities typically reported for liquid biopsy assays a Z-score of 6 to 10 should be sufficient. (under the assumption of a normal distribution, in itself a difficult assumption to make for sequencing errors, at a Z-score of 20 only 10^{-89} percent of data would fall outside of this interval. Side note: estimated number of atoms in the observable universe 10^{80})

b. Sample specific LOD. This method is most likely necessary, but whether it is sufficient remains unanswered. Many analytical approaches have used a sample specific correction as well as a nucleotide specific correction. It would be interesting to learn why the authors feel confident that a nucleotide specific correction is not needed (specifically to address the concern for overfitting by using individual cutoffs).

4) Available DNA molecules. It appears from the description in the methods section that the

authors use a general indexing strategy to allow multiplex sequencing analysis, but have not implemented a single molecule barcoding strategy (or unique molecular identifier). In our experience these molecular barcodes provide essential information for the detection of mutations and especially for the characterization of results. It would be good to learn why the authors do not see the need for individual molecule identification, especially with regards to some points:

a. Limit of detection. The authors claim to have detected mutations as low as 0.25%. In order to know how reliable this result is, it is crucial to understand how many molecules have participated in the sequencing reaction. The authors describe to sequence at 20.000x coverage, but do not explain what level of redundancy they achieve. In other words: How many DNA molecules are represented on the sequencer and how many times are they sequenced. This step is crucial to allow an assessment of the results. Unfortunately, in the liquid biopsy space many results are published that do not take into consideration the entire workflow and discard the loss of available molecules through the process. The authors clearly understand this dilemma and have shown a great deal of concern for these topics. In parts they have diligently explored some parts of this problem (analysis of copy number by input material). In order to allow the reader unbiased interpretation of the presented result is necessary to provide the copy number background.

b. Section: Cell free DNA yields, page six, line 71 onwards. This section is a little bit ambiguous and would also benefit from a more detailed description of how molecules in the plasma translate to molecules on the sequencer and how often each molecule is represented. We believe we understand the point the authors want to make, but the fact that we had to conference on it with three people to figure it out, seems to indicate that this section would benefit from some additional clarity.

5) Extension of claims. The presented work is unique and diligently executed. However, it is academic in nature. The authors make several claims about the extension into clinical practice that overstate the maturity of the current work.

a. Page 13, line 229: "novel analytical approach...", sample specific correction has been described before. The approach does not really suppress errors. It provides a sample specific cutoff to adjust for sample specific differences in error profiles

b. Use of Sensitivity and Specificity. In the clinical context these terms are typically interpreted as characteristics of a test. Here they are used to describe the presented test result. The authors should be careful in making this difference clear. The data is not yet comprehensive enough to judge sensitivity and specificity. Examples:

i. Page 13, line 231: "near perfect specificity".

ii. Page 13, line 230: "analytical sensitivity at AFs as low as 0.25%". This data is not analytical in nature. An analytical study requires a known input and comparison with a measured output. The data does not allow to judge how reliably this test will identify a 0.25% mutation. (The authors also do not follow the ACMG recommendation for analytical validation before using the test on patient samples.)

iii. Page 13, line 237: "is scalable ...substantially larger..." Scaling these ultrasensitive assays up is non trivial at best and often proves unfeasible because of complications with specificity. If the authors believe their method avoids these issues they should provide more detail.

6) Minor points:

a. Supp Fig 10, True Negative and False negative are mixed up

b. Colors in figure 4 should be optimized to allow better discrimination

c. Page 8, starting at line 118. There is likely no correlation because the contribution of MM material is different from patient to patient. The authors should explain this to the reader or focus only on the patient specific analysis, which is the more relevant.

In summary the presented work is very important and likely to directly benefit patients with MM in the future. The presented work should be strengthened in the analytical components to allow unbiased evaluation by the readership, but I believe it is a very strong paper that should find a broad interested audience.

Reviewer #4 (Remarks to the Author):

This is a well done study but it is just a little dull!

Reviewer #5 (Remarks to the Author):

The authors report on using peripheral blood to assess DNA mutational status of tumors cells from patients with MM, a disease that typically requires a more invasive BM biopsy for proper evaluation. They convincingly demonstrate that it is feasible to use the PB, with results similar to those obtained from the BM. This work is significant since efforts in the field of cancer are increasingly aimed at precision medicine. IN particular clinical trials of targeted agents that would otherwise require a BM pre-screening can now be done with PB, for which the patients will be grateful. Therefore this is more than simply a technical advance, but also a clinical advance. The manuscript is well written and and the data convincing.

Response to Reviewers' Comments

Comments from Reviewer #1 were not available.

Response to Reviewer #2

Reviewer's Comments: The manuscript from Kis et al is well timed, well written and with some revision will make an important impact in the burgeoning field of liquid biopsy. The rationale for cfDNA analysis in the management of multiple myeloma (MM) with personalised treatment approaches is clear, where replacing bone marrow aspiration will be of significant benefit to patients, who would then not have to suffer what is often a painful procedure that is not always without risk. The authors set out to make this case by assessing cfDNA in 48 MM patients sequencing all protein coding regions of 5 oncogenes relevant to this disease. The approach is not novel of course, but the application to MM is important and there are no published high quality studies of cfDNA in MM to my knowledge. Importantly, the authors compare cfDNA to matched bone marrow aspirates and report concordance of 44/46 mutations detected. A significant aspect of the manuscript is the development of the bioinformatics pipeline to deal with minimising false data. The case studies they present are also very interesting and well described. However, the authors make some overstated claims about their data which need to be revisited.

We thank the reviewer for the positive reflection on our work and this manuscript. The reviewer's main concerns revolved around tempering our interpretation of these promising, new data. Following reviewer's suggestions, we have revised these statements in the revised manuscript, as described in point-by-point response below.

1. *They have not incorporated germline analysis that most in the field of cfDNA would consider mandatory. Without this built into their workflow, however good their informatics may be, it is not possible to say with complete surety that they pick up somatic mutations. They should either provide and incorporate germline data into their analysis or refer throughout their manuscript to potential somatic mutations.*

We agree with the reviewer that mutations identified by our analysis should be referred to as "likely somatic" rather than unqualified "somatic", since we used population data to classify sequence variants as non-pathogenic germline single nucleotide polymorphisms (SNPs) instead of using direct sequence analysis of germline DNA in each patient. At the onset of this study, we chose not to sequence genomic DNA in all patients enrolled in the study, as this was not routinely done by the clinical laboratory responsible for bone

marrow testing at our centre. In addition to remaining consistent with current practice, we also sought to lower the cost of sequencing required for each patient to facilitate future clinical uptake of this method for inexpensive screening of patients for enrolment onto clinical trials and/or serial monitoring of the identified mutations in patients during the course of therapy. Given the availability of many large databases of population sequencing data (e.g., dbSNP, 1000Genomes, ExAC, and ClinVar), we felt that we could accurately identify the vast proportion of germline SNPs in each patient by crosschecking them against the list of validated germline SNPs identified in the general population. Certainly, this approach is dependent on up-to-date databases that capture germline variation reflective of the patient population and there is a possibility that rare germline variants could be missed by our current bioinformatics pipeline, particularly as target panels increase in size. Given this early decision, we did not bank matched normal material at the time of analysis for every patient. In light of this reviewer's feedback, we are now banking all buffy coat cells from blood for all studies going forward.

To acknowledge this issue in the revised manuscript, we have noted this possibility in the updated Methods text (shown below). Furthermore, as suggested by the reviewer, in the revised manuscript and supplementary methods, we now refer to all mutations identified by our algorithm as "likely somatic" rather than "somatic".

"While this approach allowed us to effectively differentiate between germline SNPs and tumor-derived somatic variants without matching normal DNA, it is possible that rare germline variants not represented in the publically available population variation databases may not be appropriately filtered by our bioinformatics pipeline. When interpreting data for patients with evidence of high-AF mutation calls (>40%) or applying this method to a larger target panel, it may be necessary to test for the presence of potential germline variants through sequencing of matched DNA obtained from leukocytes readily available from the same blood samples used for cfDNA extraction." (Page 20, Methods section)

2. *The authors refer to the inter-patient variability in ctDNA yields that they observe as unusual but this is just as likely to reflect a relatively low sample number and others have reported a similar level of variability, which the authors should refer to. e.g., Establishment of tumor-specific copy number alterations from plasma DNA of patients with cancer. Heitzer et al Int J Cancer. 2013.*

We thank the reviewer for this suggestion. In the revised manuscript, we have toned down the statement made about the inter-patient variability in cfDNA yields and cited the reference publication suggested by the reviewer. Now added to Page 5, Results section:

"High inter-patient variability in cfDNA yields among patients with MM (4.7-331 ng/mL plasma) and other malignancies (0-365 ng/mL plasma) observed in this study was consistent with other studies of patients with advanced cancers.¹"

3. *The allele frequency (AFs) the authors report are also not "unusually" high. e.g. Noninvasive Identification and Monitoring of Cancer Mutations by Targeted Deep Sequencing of Plasma DNA. Forsheo et al. Science Translational Medicine 2012.*

We agree with the reviewer and have fixed this statement in the revised manuscript, Page 15, Discussion section:

“Increased levels of ctDNA in advanced stage MM may also explain relatively high AFs of tumor-derived mutations (11% average, 0.25-46% range) measured by LB-Seq in this study population. Although comparable cfDNA yields and mutant AFs have been reported in other advanced tumors,¹⁻³ some cancer types may have lower yields of cfDNA and lower AFs for tumor-specific mutations detectable by LB-Seq. This may affect the feasibility of utilizing LB-Seq to achieve the comparable sensitivity in other patient populations or necessitate collection of increased volumes of plasma.”

4. *The method the authors use for targeted sequencing is not really that novel and other publications have also sequenced all protein coding exons of genes, e.g., Tumor-associated copy number changes in the circulation of patients with prostate cancer identified through whole-genome sequencing. Heitzer et al Genome Medicine 2013 and High-Throughput Detection of Actionable Genomic Alterations in Clinical Tumor Samples by Targeted, Massively Parallel Sequencing, Wagle et al Cancer Discovery 2012.*

As the reviewer correctly points out, targeted sequencing of all protein coding exons of genes has been reported previously for genetic profiling of solid tumors as well as cfDNA in prostate cancer. However, the use of this approach for cfDNA sequence analysis is rare and we are the first to report on its application in multiple myeloma. We have clarified this in the Discussion section on Page 15:

“Rather than looking only at mutation hotspots, we targeted all protein-coding exons of genes of interest to allow, in a single assay, simultaneous detection and tracking of any mutations in the target region over the course of therapy^{4,5}. This is of importance in MM, a disease characterized by an abundance of subclonal genetic abnormalities (including adjacent mutations affecting the same codon within distinct subclones), high degree of intra- and inter-patient genetic heterogeneity, and treatment-associated secondary resistance mutations.⁶⁻⁹”

5. *Finally, the authors should comment further on the scalability of their approach, adding detail on the maintenance of sensitivity as the gene panel under study expands.*

We agree that discussion of the scalability of this method is an important aspect of that should be addressed in more detail in the paper. Based on previously published studies describing cfDNA sequencing for whole exome and whole genome analysis as well as our own recent work in myeloma and other cancer types using large targeted panels, we expect our approach to scale well with increasing number of baits. As suggested by the reviewer, we have revised the Discussion section on page 16, to elaborate further on the scalability of this method, in particular as it pertains to the maintenance of sensitivity of this approach and the need for germline controls as the panel size increases.

“While we included only five genes in this study, we expect that LB-Seq is scalable for examination of larger portions of the genome to query other target genes or mutation classes (such as rearrangements and copy number alterations), as proof-of-concept studies of cfDNA exome and genome sequencing have been reported in other disease sites^{2,5}. With increasing panel size, the sensitivity of this approach may be affected by poor mapping quality in specific genomic regions (e.g., pseudogenes, high homology regions), poor capture efficiency or coverage uniformity, increased sequence bias, and increased cost of sequencing necessary to achieve a similar depth of coverage.

Some of these challenges may be overcome by parallel sequencing of matched normal DNA and further improvement of library preparation and target capture methodology to minimize polymerase errors and sequencing artefacts (e.g., applying duplex sequencing method with molecular barcoding).^{10,11}

With the text revisions above resolved, this manuscript is worthy of publication in Nat Comms in my opinion.

We are grateful for this positive feedback and useful suggested revision that helped improve the quality of our manuscript.

Response to Reviewer #3

Reviewer's Comments: The manuscript "Circulating tumor DNA sequences as an alternative to multiple myeloma bone marrow aspirates" by Kis et al. compared mutation profiles detected in bone marrow to those that can be established from circulating cell free DNA. After setting up the methodology by using a training and validation set, the authors find that cfDNA analysis in most cases accurately reflects the mutational status detected by analysis of the bone marrow. Overall the authors conclude that cfDNA analysis provides a reasonable alternative to bone marrow aspirates.

The manuscript is very well written and clearly shows what a remarkable effort has gone into this work. There are a number of minor and major points that should be addressed before publication.

We thank the reviewer for this positive feedback on our research study and a thorough review of this manuscript and have prepared a detailed response to the major and minor points identified by the reviewer, described below.

Major Points:

1. *Clarity of the methodology. The authors do an outstanding job in being transparent about their work. In the supplemental information they provide data as well as analytical scripts so their work can be replicated if so desired. The challenge to date is that given the current limitations on review time and resources a comprehensive validation of these techniques is practically impossible. Ultra sensitive detection of DNA mutations from cfDNA by using next generation sequencing is not a widely adapted method today and still in its infancy. Many analytical errors that can be made are hard to uncover, but have severe consequences on the results. Since it is unlikely that we will see a shift in the review process for biological papers, that more closely resembles that review of mathematical papers anytime soon, I recommend that the authors provide a bit more detail in the main text about the analytical methods, alternatively provide a more detailed write-up about the rationale behind their analytical steps. In its current form it is hard to judge the quality of the analytical pipeline. On the surface some items are cause for concern:*
 - a. *Use of MuTect as primary variant identification. The chosen setting for MuTect appear to be correct, but MuTect traditionally has underperformed in settings where sub 5% mutations need to be detected (compared to specialized solutions)*

We agree with the reviewer that the manuscript would benefit from a more thorough explanation regarding our choice to use MuTect for analysis of rare variants in deep-sequencing cfDNA data, which is not a default setting for this tool. In this study, we explored the utility of using muTect for analysis of low frequency mutations in cfDNA, which required adjustment of several settings for this analysis as well as downstream filtering of the resulting variant calls using muTect-generated parameters. Due to the lack of published data on the application of muTect for this analysis, settings used for this analysis had to be determined experimentally using a training cohort of 25 cfDNA samples with matching tumor sequencing data. Furthermore, some changes made to the settings introduced new sources of error calls that were now passing the muTect filtering process that we had to remove in the downstream analysis. One such example relates to allowing multiple variants to be called at the same genomic position, which lead to overestimation of the allele fractions at homozygous SNP loci (described in more detail in the answer to the next Question #1b). We have addressed this issue in the Methods section on page 19 of the revised manuscript:

“In this study, we configured muTect for analysis of low frequency mutations in cfDNA by setting “fraction_contamination” to 0 to enable emission of low frequency variants, setting “gap_events_threshold” to 1000 to increase the number of reads tolerated with indels in high coverage data, and enabling the “force_alleles” flag to prevent rejection of mutations due to triallelic sites (see details in the Supplementary Methods section).”

The updated Supplementary Methods section now reads as follows:

Detection of low frequency mutations in cfDNA deep sequencing data

When we initiated this study, bioinformatics tools for discovery of ultra-rare mutations (with allele fractions < 1%) in cfDNA deep sequencing data (>20,000X) were lacking. MuTect is a widely used bioinformatics tool for calling single nucleotide variants (SNVs) and is currently recommended by the Genome Analysis Toolkit Best Practices document for mutation detection in exome, whole genome, and targeted panel sequencing studies (<https://software.broadinstitute.org/gatk/best-practices/>). In our study, we explored the utility of using muTect for analysis of low frequency (<1%) mutations in ultra-deep cfDNA sequencing data (>20,000X) without matching normal DNA using adjusted configuration. The configuration changes we implemented included reducing “fraction_contamination” threshold to 0, increasing “gap_events_threshold” to 1000 due to increased target coverage in cfDNA data, as well as using “force_alleles” argument to prevent rejection of mutations due to triallelic sites (**Supplementary Figs. 6B and 7**). ” (Page 37, Supplementary Methods)

- b. *Please clarify the first step in the R filtering described in the supplement:” Remove mutation calls rejected by MuTect or supported by less than 10% of total reads at that locus”. It currently reads as if mutations are rejected that have less than 10% of all reads at this position showing the mutation, which in turn makes it hard to understand how mutations in the sub 10% are detected without the use of single molecular barcodes. Maybe this is a simple typo and it should read “10 total reads”*

We apologize for the lack of clarity in regards to this initial filtering step in our algorithm and agree that a better explanation needs to be provided. As a follow up to the

discussion provided for the previous question from this reviewer (Question #1a), in order to use muTect to call low AF variants in cfDNA, we had to change several analysis parameters, which also opened up some new issues that needed to be resolved in downstream filtering. One such issue was related to using “force_alleles” argument during muTect analysis. The default settings of muTect allow only one variant with the highest tumor LOD score to be called at any particular locus. Forcing alleles at tri-allelic sites allows detection of up to three independent single-base substitutions (e.g., A>C, A>G, and A>T) at the same genomic locus. This modification was important for cfDNA data analysis since overlapping mutations at the same locus may originate from distinct tumor DNA fragments secreted from different tumor sites in the body. This is particularly important for MM, as distinct subclones may have alternate mutations at the same DNA position resulting in different amino acid substitutions.

As suggested by the reviewer, we clarified this section on pages 37-38 of the Supplementary Methods, as shown below.

“Some of the changes made in muTect parameters generated new analysis issues that needed to be resolved in downstream filtering. One such issue was related to using “force_alleles” argument during muTect analysis. The default settings allow only one variant with the highest tumor LOD score to be called at any particular locus. Forcing alleles at triallelic sites allows detection of up to three independent single-base substitutions (e.g., A>C, A>G, and A>T) at the same genomic locus. This is important for cfDNA data analysis since overlapping mutations at the same locus may originate from distinct tumor DNA fragments derived from different clonal populations. However, this adjustment led to some false mutation calls with apparently high AFs at loci where homozygous, non-reference sequence SNPs were present. Because 90-99.9% of the reads at these loci typically corresponded to the non-reference allele (i.e., homozygous SNPs), the reference allele count can be very low. MuTect analysis calculates tumor_f values independently for each substitution ($tumor_f = \frac{t_alt_count}{t_ref_count + t_alt_count}$), with denominator being the sum of alternative and reference allele counts and not the total read count at that locus. At homozygous SNP loci, this leads to overestimation of the AFs for the remaining two possible single-base substitutions and results in uncharacteristically high tumor_f and tumor LOD scores (dependent on tumor_f) for these variant calls. To avoid calling these erroneous calls as real mutations, we have removed any mutations for which the sum of reference and alternative allele counts equals to < 10% of total reads. This threshold was selected based on the assumption that at homozygous SNP loci, > 90% of the reads will correspond to the alternative allele, leaving <10% of reads for the reference allele and the other two possible alternative variants. These mutation calls were removed prior to converting the tumor LOD scores into the modified Z-scores to avoid skewing the LOD-score distribution using these low confidence error calls with apparently high LOD scores.”

2. *Training and Validation. The authors choose a common model including training, test and validation sets.*
 - a. *Supplement Figure 8 shows the use of 59 specimen analyzed in these three phases. The 59 specimen are derived from 48 samples. A total of 10 samples contributed more than one specimen. These specimens are no longer independent and hence their information to some level redundant (from figure 4 it appears at least 4 to 6 samples are affected in the validation*

set). For example, if all 10 samples, with more than one specimen available, are part of the 16 specimen used in the “without BM” set, the set would no longer qualify as truly independent, because a major subset of specimen had been used to train on. This point should be clarified, ideally the analysis should be repeated without the use of multiple aliquots from one sample. Even though the numbers will be smaller, it is likely that the results will be similar.

This is a valid concern raised by the reviewer, and we agree that our original validation cohort may not have been completely independent because it includes 4 serial samples (namely MYL-001(2), MYL-012(2), MYL-022(2), and MYL-027(2)) from patients whose previous cfDNA samples were included as part of the training cohort. To compensate for the removal of these samples, we have added data from 5 new patients to the validation cohort that we have subsequently tested. Following reviewer’s suggestion, we have repeated the analysis of the sensitivity and specificity in the validation set omitting the data from the four samples that overlap with the training cohort. We have made this explicitly clear in the main text and an updated Supplementary Figure as follows:

“We then validated our method using a blinded validation cohort of 19 cfDNA samples from 17 patients with matching BM sequencing data (**Supplementary Fig. 8**). Four additional samples included in the validation set were serial cfDNA specimens from patients included in the training cohort (**Fig. 3, ***) and, hence, were excluded from validation of the Z-score threshold but were included in the overall calculations of sensitivity and specificity across all samples analyzed in this study.” (Page 7, Results section)

Supplementary Figure 10

Supplementary Figure 10. Sensitivity and specificity of cfDNA sequence analysis in training and validation cohorts. The training cohort consisted of 25 cfDNA samples from 23 MM patients with 25 matching BM-derived tumor DNA samples collected within 1 week of blood draw. The validation cohort included 19 matched cfDNA and BM-derived DNA samples from 17 additional patients. We also analyzed four serial cfDNA samples from patients in the training cohort that were excluded from validation of the Z-score threshold but were included in the overall calculations of sensitivity and specificity across all samples analyzed in this study (bottom diagram).

- b. *Batch effect for Tumor LOD scores. Figure 4 shows the LOD scores for all 59 specimens. There is a clear batch effect displayed between the three groups. The baseline in the training model appears to be lower compared to the testing populations. It would be necessary to explain the reasons for the observed effect and why it is not a cause for concern.*

With the inclusion of additional data, we continue to see fluctuations in tumour LOD score distributions across runs, illustrating the robustness of our approach to compensate for variability in sample, library, and sequencing quality encountered during routine clinical testing. We agree that this topic warrants further discussion in the manuscript and we have added a new paragraph discussing the batch effect on sequencing data quality in the Results section.

“Batch effect on sequencing data quality

We identified a batch effect on sequencing data quality – the number of variants called in each sample and the absolute LOD scores obtained from MuTect analysis. This batch dependence may be related to several factors such as increased sequencer error rate due to equipment variation, increased DNA damage during overnight hybridization, sample processing, or storage due to changes in reagents, experimental conditions, or personnel.¹² In particular, two batches of samples for which target capture was done on the same day and that were sequenced in the same sequencing run, had higher absolute LOD scores (median: 23.2; range: 18.5-26.9) compared to the previously-sequenced training cohort (median: 8.5; range: 6.5-16.5). Coincidentally, 16 of 23 cfDNA samples in the validation cohort were included in the two batches with higher LOD scores (**Fig. 4**). Despite this potential batch effect, the LB-Seq analysis pipeline was able to effectively differentiate between real genetic variants and sequencing artifacts in each sample by using sample-specific LOD score thresholds. This resilience against batch effect and variation in data quality is likely to be beneficial for implementing this method in other research laboratories and in clinical setting.” (Pages 8-9, Results section)

Figure 4

Figure 4. Distribution of tumor LOD scores in cfDNA sequencing data. For each sample, all candidate mutation calls generated by muTect version 1.1.4 were divided into subgroups based on the type of mutation or the filtering step at which they were removed, as indicated in the legend. Stripcharts showing the tumor LOD scores for each subgroup of mutations were overlaid with the boxplot demonstrating the distribution of tumor LOD scores for all mutation calls kept by muTect within each sample, prior to downstream filtering. The central rectangle spans the first to the third quartile (IQR), and segment inside the rectangle shows the median LOD score for each sample. Sample-specific thresholds for calling somatic mutations (-) were determined as the tumor LOD score corresponding to the modified Z-score of 20 (i.e., 20 MADs above the median). Unfiltered annotated data for all samples are available as **Supplementary Data File 2**. Custom Rscript (in R version 3.2.2) for filtering and plotting LOD score distribution data is available at www.github.com/pughlab/lb-seq.

3. Analysis.
 - a. The use of 20 as a cutoff for the modified Z-scores appears to be derived from empirical data. There is a certain risk for overfitting (see above), consequently it would be good to explain why a Z-score of 20 is justifiable from a theoretical basis or why the standard model does not apply. To achieve specificity High specificities typically reported for liquid biopsy assays a Z-score of 6 to 10 should be sufficient. (under the assumption of a normal distribution, in itself a difficult assumption to make for sequencing errors, at a Z-score of 20 only 10^{-89} percent of data would fall outside of this interval. Side note: estimated number of atoms in the observable universe 10^{80})

We thank the reviewer for this interesting analysis and opportunity to explain our approach to data analysis. While we agree that in other liquid biopsy assays, lower Z-scores of 6-10 may be sufficient for identification of real variants, in the case of using the distribution of tumor LOD scores to identify real genetic variants from sequencer errors, choosing a Z-score in this range as a threshold was not adequate. The primary reason for this is that the tumor LOD scores are not normally distributed. Instead, their distribution is highly skewed, with hundreds or thousands of sequencer errors that have low LOD scores (median tumour LOD score 6 - 30, depending on a sample/batch) and only a few real genetic variants (somatic mutations and germline SNPs) that have high or extremely high LOD scores (ranging from 100 to 100,000). This skewed distribution can be observed in Figure 4. The boxplots represent the majority of LOD scores in each sample, corresponding to sequencing artifacts, and the blue and red dots represent likely somatic variants and germline SNPs, respectively. The LOD scores are plotted on a log scale and span several magnitudes, further highlighting the skew in these data.

For this reason, we applied the modified Z-score transformation, a method independent of normality assumption, which relies on median and median absolute deviation for LOD score distribution in each sample. Since this approach has not been used previously to transform tumor LOD scores distribution data, we decided to establish the threshold value for the modified Z-score empirically through plotting the Receiver Operator Curve for the 25 cfDNA samples with matching bone marrow tumor DNA sequencing data available (i.e., the training cohort). This ROC analysis is presented as the Supplementary Figure 9. The modified Z-score of 20 achieved optimal sensitivity (26 of 27 mutations detected in the bone marrow DNA samples were correctly identified in the matching cfDNA) and very good specificity (only one mutation not detected in the bone marrow was reported, although we believe this to be a real mutation, as discussed in the paper). As an example, if the modified Z-score used to call real variants in this training dataset was lowered to 10, the number of false positives would increase from 1 to 27, and 201 false positives would be called using 6 as the threshold value. We strongly feel that the empirical approach we used to identify the threshold value (i.e., using the ROC curve with the training cohort of 25 samples) was more appropriate in this situation.

“To account for variability of the absolute tumor LOD score values between samples and sequencing batches, tumor LOD scores were transformed into modified Z-scores, i.e. the distance from the median divided by the median absolute deviation (MAD)¹³ for each sample, which are independent of the normality assumption. Similar approach has been described previously for transformation of high throughput screening data generated for large-scale RNA interference libraries.¹⁴ Since the use of this approach for transforming tumor LOD score distribution data has not been described previously, we used the receiver-operator curve (ROC) generated using the training cohort of 25 cfDNA samples with matching BM data available to establish the threshold value for the modified Z-score, which would allow us to differentiate true mutations from sequencing artifacts (**Supplementary Fig. 9**). From ROC analysis, the modified Z-score of 20 (i.e., tumor LOD score 20 MADs above the median LOD score in each sample) achieved optimal sensitivity and specificity and was used to identify bona fide variants in all other samples.” (Page 20, Methods section)

- b. *Sample specific LOD. This method is most likely necessary, but whether it is sufficient remains unanswered. Many analytical approaches have used a sample specific correction as well as a*

nucleotide specific correction. It would be interesting to learn why the authors feel confident that a nucleotide specific correction is not needed (specifically to address the concern for overfitting by using individual cutoffs).

We agree with the reviewer that nucleotide correction is indeed important and it has been applied at early stages in our bioinformatics pipeline. GATK Best Practices currently involves base quality recalibration that was completed for all samples. Furthermore, muTect analysis also looks at sequence context analysis for each variant call and incorporates this information into calculation of the tumor LOD scores. Furthermore, tumor LOD scores are calculated in the forward and reverse directions. We used this information to identify sequence-context dependent systematic sequencer errors (if the LOD score in the forward and reverse directions were more than 5-fold different). The use of the sample specific LOD score threshold was the last step of our filtering process used to remove sequencing artifacts that were missed by all other filtering approaches.

4. *Available DNA molecules. It appears from the description in the methods section that the authors use a general indexing strategy to allow multiplex sequencing analysis, but have not implemented a single molecule barcoding strategy (or unique molecular identifier). In our experience these molecular barcodes provide essential information for the detection of mutations and especially for the characterization of results. It would be good to learn why the authors do not see the need for individual molecule identification, especially with regards to some points:*
 - a. *Limit of detection. The authors claim to have detected mutations as low as 0.25%. In order to know how reliable this result is, it is crucial to understand how many molecules have participated in the sequencing reaction. The authors describe to sequence at 20,000x coverage, but do not explain what level of redundancy they achieve. In other words: How many DNA molecules are represented on the sequencer and how many times are they sequenced. This step is crucial to allow an assessment of the results. Unfortunately, in the liquid biopsy space many results are published that do not take into consideration the entire workflow and discard the loss of available molecules through the process. The authors clearly understand this dilemma and have shown a great deal of concern for these topics. In parts they have diligently explored some parts of this problem (analysis of copy number by input material). In order to allow the reader unbiased interpretation of the presented result is necessary to provide the copy number background.*

This is a great question and we can attempt to estimate the duplication rate and the number of unique cfDNA molecules that make it through to sequencing. Ultimately, we are now implementing molecular barcoding techniques to measure library diversity more directly. However, these methods were not available when we began this study. As suggested by the reviewer, we have revised the section on cfDNA yields to provide a more detailed description of how molecules in the plasma translate to molecules on the sequencer.

“As described in detail in the Supplementary Methods section, we estimated that for samples with 83ng of cfDNA used to prepare sequencing libraries (approximately 4.3×10^{11} unique cfDNA fragments), a median 43% (1.8×10^{11}) of fragments were retained following adapter-ligation, library amplification, and bead clean up. Due to having a small target region (17.5 kb), only a small fraction of these molecules correspond to the region of interest, providing an estimated 1.1

$\times 10^6$ unique and usable cfDNA fragments available for capture. This corresponds to unique cfDNA fragment coverage of approximately 11,000X for a region with 20,000X coverage (~55%). This percentage will decrease in regions with higher coverage (~11% unique reads for regions with 100,000X coverage).” (Page 6, Results section)

“Reads were aligned to the human genome reference (Illumina iGenomes hg19) using bwa¹⁵ version 0.7.12 and post-processed following Genome Analysis Toolkit (GATK) Best Practices¹⁶ without marked duplicates, to avoid mistakenly reducing the true library complexity, as described in Supplementary Methods and CAPP-Seq protocol.¹⁷” (Page 19, Methods section)

We have also added a section in the Supplementary methods addressing this question in further detail:

“Estimation of duplication rate and loss of unique cfDNA molecules during sample processing

For a typical cfDNA sample, 83 ng of cfDNA was used to prepare adapter-ligated sequencing library. Since cfDNA is naturally fragmented with a major fragment size corresponding to a DNA loop around one nucleosome (~166 bp, 80-85%) and a second peak (~15-20%) corresponding to two nucleosomes (332 bp), we estimated the average fragment size of cfDNA to be approximately 180 bp. Using 650 Daltons as an average weight of one DNA base pair, we estimated that 83 ng of cfDNA contains approximately 4.3×10^{11} unique DNA fragments (7.1×10^{-13} mols). In this study, the estimated efficiency of cfDNA ligation to NEXTflex-96 adapters for libraries prepared using 83 ng of input DNA and four amplification cycles (n=33) was median 43% (21-70% range, **Supplementary Fig. 3b**). Hence, the estimated number of unique cfDNA molecules converted into a barcoded DNA-seq library is 1.8×10^{11} ($0.88-3.0 \times 10^{11}$). Since the target region (146 x 120 bp probes = 17,5 kbp) makes up only 0.00057% of the genome, we estimated that approximately 0.00057% of cfDNA molecules present in each sample will align to the region of interest and will be captured for sequencing (1.1×10^6 unique and usable cfDNA fragments). Assuming 180 bp to be the average size of cfDNA fragments, approximately 98 fragments will be required to achieve 1X coverage of the 17.5 kbp target region. Hence, we estimate that the median depth of coverage that can be achieved through sequencing of 1.1×10^6 unique cfDNA fragments is approximately 11,000X. However, it will depend on the efficiency of ligation and post-ligation PCR during library preparation for each sample, as well as PCR efficiency differences for different cfDNA fragments.

If all of the above assumptions hold true, mutations detected in a region with 20,000X coverage should have ~55% unique reads, and this fraction will decrease in regions with higher coverage (~11% unique reads for regions with 100,000X coverage). Higher degree of duplication is likely to be present in samples with lower initial cfDNA input into library constructions or in libraries prepared using Illumina LT adapters (the ligation efficiency was lower compared to NEXTflex adapters, **Supplementary Fig. 3**). If the duplication rate is similar between normal and tumor-derived cfDNA fragments during sample processing, the AFs observed for tumor-derived mutations remain largely independent of the duplication rate and representative of the allele fractions present in plasma.” (Pages 35-36, Supplementary Methods)

- b. *Section: Cell free DNA yields, page six, line 71 onwards. This section is a little bit ambiguous and would also benefit from a more detailed description of how molecules in the plasma translate to molecules on the sequencer and how often each molecule is represented. We believe we understand the point the authors want to make, but the fact that we had to conference on it with three people to figure it out, seems to indicate that this section would benefit from some additional clarity.*

We thank the reviewer for a great recommendation. We have included a more detailed discussion of how cfDNA molecules extracted from plasma translate to unique cfDNA fragments available for sequencing, as discussed above in our response to Question 4a from this reviewer. We have now added a section on Page 6 for the Results section addressing this issue:

“As described in detail in the Supplementary Methods section, we estimated that for samples with 83ng of cfDNA used to prepare sequencing libraries (approximately 4.3×10^{11} unique cfDNA fragments), a median 43% (1.8×10^{11}) of fragments were retained following adapter-ligation, library amplification, and bead clean up. Due to having a small target region (17.5 kb), only a small fraction of these molecules correspond to the region of interest, providing an estimated 1.1×10^6 unique and usable cfDNA fragments available for capture. This corresponds to unique cfDNA fragment coverage of approximately 11,000X for a region with 20,000X coverage (~55%). This percentage will decrease in regions with higher coverage (~11% unique reads for regions with 100,000X coverage).”

5. *Extension of claims. The presented work is unique and diligently executed. However, it is academic in nature. The authors make several claims about the extension into clinical practice that overstate the maturity of the current work.*
- a. *Page 13, line 229: “novel analytical approach...”, sample specific correction has been described before. The approach does not really suppress errors. It provides a sample specific cutoff to adjust for sample specific differences in error profiles*

As described in our answer for this Reviewer’s Question #3a, the analysis pipeline implemented in this method relies on using the modified Z-score analysis to transform the absolute LOD scores (a muText log odds output parameter) using the median and median absolute deviation (MAD) for the distribution of tumor LOD scores in each sample, which, to the best of our knowledge, has not been described previously for ctDNA analysis. Since the tumor LOD scores are not normally distributed, the benefit of using the modified Z-scores is in its independence of the normality assumption (unlike mean and standard deviation used to calculate traditional Z-scores). The distribution of LOD scores is highly skewed, with hundreds or thousands of sequencer errors that have low LOD scores (median LOD score 6 - 30, depending on the sample/batch) and only a few real genetic variants (somatic mutations and germline SNPs) that have high or extremely high LOD scores (100 - 100,000). This type of approach has been successfully applied to the high throughput screening of large-scale RNA interference libraries¹⁴ and we are the first to apply this type of analysis to deep-sequencing genomics data or LOD score distribution data. However, we have toned down this statement and now refer to it as “a new analytic approach” to avoid the extension of claim on the intrinsic novelty of this method. We have modified the corresponding statement in the Discussion:

“We also implemented and validated a new analytic approach for analysis of cfDNA sequencing data that allowed detection of tumor-derived mutations in cfDNA with AFs as low as 0.25% with 96% sensitivity and >98% specificity, as compared to the state-of-the molecular profiling of the BM tumors.” (Page 15, Discussion section)

“To account for variability of the absolute tumor LOD score values between samples and sequencing batches, tumor LOD scores were transformed into modified Z-scores, i.e. the distance from the median divided by the median absolute deviation (MAD)¹³ for each sample, which are independent of the normality assumption. Similar approach has been described previously for transformation of high throughput screening data generated for large-scale RNA interference libraries.¹⁴ Since the use of this approach for transforming tumor LOD score distribution data has not been described previously, we used the receiver-operator curve (ROC) generated using the training cohort of 25 cfDNA samples with matching BM data available to establish the threshold value for the modified Z-score, which would allow us to differentiate true mutations from sequencing artifacts (**Supplementary Fig. 9**). From ROC analysis, the modified Z-score of 20 (i.e., tumor LOD score 20 MADs above the median LOD score in each sample) achieved optimal sensitivity and specificity and was used to identify bona fide variants in all other samples.” (Page 20, Methods section)

- b. *Use of Sensitivity and Specificity. In the clinical context these terms are typically interpreted as characteristics of a test. Here they are used to describe the presented test result. The authors should be careful in making this difference clear. The data is not yet comprehensive enough to judge sensitivity and specificity. Examples:*
- i. *Page 13, line 231: “near perfect specificity”.*
 - ii. *Page 13, line 230: “analytical sensitivity at AFs as low as 0.25%”. This data is not analytical in nature. An analytical study requires a known input and comparison with a measured output. The data does not allow to judge how reliably this test will identify a 0.25% mutation. (The authors also do not follow the ACMG recommendation for analytical validation before using the test on patient samples.)*

The reviewer raised some valid concerns about the use of sensitivity or specificity to describe the validation of this test using BM sequencing data, and we understand that this terminology may lead to perception that we have completed an independent analytical validation of this test on patient samples. However, this is also the terminology, which is commonly used to present test results (e.g., in contingency table or ROC analysis). We feel it is not possible to completely avoid the use of terms “sensitivity” or “specificity” in this paper and still convey the message of test validation using training and validation cohorts. To aid with an accurate interpretation of the significance of our findings, we have clarified the meaning of these terms in the context of test result interpretation presented in this manuscript and have toned down the language used to describe these terms (i.e., “near perfect specificity”, “analytical sensitivity at AFs as low as 0.25%”).

“We also implemented and validated a new analytic approach for analysis of cfDNA sequencing data that allowed detection of tumor-derived mutations in cfDNA with AFs as low as 0.25% with 96% sensitivity and >98% specificity, as

compared to the state-of-the molecular profiling of the BM tumors.” (Page 15, Discussion section)

iii. Page 13, line 237:” is scalable ...substantially larger...” Scaling these ultrasensitive assays up is non trivial at best and often proves unfeasible because of complications with specificity. If the authors believe their method avoids these issues they should provide more detail.

Reviewer #2 posed a similar question, to which we have provided a detailed response and have made revisions to the discussion section. Please refer to our response to Question #5 from the Reviewer #2 on Pages 3-4 of this document.

6. Minor points:

a. In Supp Fig 10, True Negative and False negative are mixed up

We thank the reviewer for pointing out this error. We have corrected the Supplementary Figure 10 accordingly. This figure is included in our response to Question #2a from this reviewer (Page 9 of this document).

b. Colors in figure 4 should be optimized to allow better discrimination

This is a great suggestion. We have revised this figure to make the colors easier to differentiate. The revised Figure 4 is shown below.

Figure 4

Figure 4. Distribution of tumor LOD scores in cfDNA sequencing data. For each sample, all candidate mutation calls generated by muTect version 1.1.4 were divided into subgroups based on the type of mutation or the filtering step at which they were removed, as indicated in the legend. Stripcharts showing the tumor LOD scores for each subgroup of mutations were overlaid with the boxplot demonstrating the distribution of tumor LOD scores for all mutation calls kept by muTect within each sample, prior to downstream filtering. The central rectangle spans the first to the third quartile (IQR), and segment inside the rectangle shows the median LOD score for each sample. Sample-specific thresholds for calling somatic mutations (-) were determined as the tumor LOD score corresponding to the modified Z-score of 20 (i.e., 20 MADs above the median). Unfiltered annotated data for all samples are available as **Supplementary Data File 2**. Custom Rscript (in R version 3.2.2) for filtering and plotting LOD score distribution data is available at www.github.com/pughlab/lb-seq.

- c. *Page 8, starting at line 118. There is likely no correlation because the contribution of MM material is different from patient to patient. The authors should explain this to the reader or focus only on the patient specific analysis, which is the more relevant.*

We agree with the reviewer that this section would benefit from a better explanation of this apparent lack of correlation in cfDNA and BM allele fractions, which we had previously omitted. Indeed, this was the reason why we decided to look more closely at patient-specific correlations between cfDNA and BM allele fractions for patient with multiple mutations. We have revised this section in the paper as shown below.

“Without grouping data by patient, AFs in cfDNA and BM did not correlate ($R^2 = 0.34$, **Supplementary Fig. 13**), likely due to differences in the relative contribution of tumor DNA to the total cfDNA in plasma of each patient. To test this hypothesis, we examined whether the clonal frequencies of mutations in BM were recapitulated in circulating tumor DNA (ctDNA) in eleven patients with multiple mutations (**Fig. 5** and **Supplementary Figs. 14 and 15**). Three of these patients had two serial cfDNA and BM samples each. Overall, mutant AFs were highly concordant between cfDNA and BM (**Fig. 5a**, R^2 range 0.913 to 0.997).” (Page 9-10, Results section)

In summary, the presented work is very important and likely to directly benefit patients with MM in the future. The presented work should be strengthened in the analytical components to allow unbiased evaluation by the readership, but I believe it is a very strong paper that should find a broad interested audience.

We thank the reviewer for a very thorough evaluation of this manuscript and this positive feedback.

Response to Reviewer #4

Reviewer's Comments: This is a well done study but it is just a little dull!

We believe that our study is exciting as it makes an important contribution to laboratory and clinical science. Our method demonstrates potential for switching to non-invasive genetic testing in MM, to replace clinically unnecessary bone marrow aspiration. We

foresee a range of other future clinical applications of this method for clinical management of multiple myeloma and other malignancies, screening for mutations required to enroll patients onto clinical trials, serial longitudinal monitoring of identified mutations during the course of therapy, and detection of minimal residual disease.

Response to Reviewer #5

Reviewer's Comments: The authors report on using peripheral blood to assess DNA mutational status of tumors cells from patients with MM, a disease that typically requires a more invasive BM biopsy for proper evaluation. They convincingly demonstrate that it is feasible to use the PB, with results similar to those obtained from the BM. This work is significant since efforts in the field of cancer are increasingly aimed at precision medicine. In particular clinical trials of targeted agents that would otherwise require a BM pre-screening can now be done with PB, for which the patients will be grateful. Therefore, this is more than simply a technical advance, but also a clinical advance. The manuscript is well written and the data convincing.

This reviewer had several positive comments about the significance of our work and the quality of our manuscript. The reviewer highlighted the significance of this work considering that “*efforts in the field of cancer are increasingly aimed at precision medicine*” especially in multiple myeloma where “*trials of targeted agents that would otherwise require a BM pre-screening can now be done with [plasma biopsy], for which the patients will be grateful.*” We fully agree with this statement and would like to add that in our experience, many patients refuse to enrol in clinical trials that require BM aspiration or cannot be properly assessed because of suboptimal quality of the BM aspirate. In some cases, we have had to repeat the BM aspiration to assess patient’s eligibility for the trial. Based on the data presented in this study, 13 of 13 patients could have been correctly stratified for enrolment in the trial using plasma aliquot alone.

We appreciate the positive feedback on the quality of our manuscript. We thank the reviewer for this assessment and agree that this method can change clinical practice in multiple myeloma.

References

1. Heitzer, E. *et al.* Establishment of tumor-specific copy number alterations from plasma DNA of patients with cancer. *Int. J. Cancer J. Int. Cancer* **133**, 346–356 (2013).
2. Forshew, T. *et al.* Noninvasive identification and monitoring of cancer mutations by targeted deep sequencing of plasma DNA. *Sci. Transl. Med.* **4**, 136ra68 (2012).
3. Bettgowda, C. *et al.* Detection of circulating tumor DNA in early- and late-stage human malignancies. *Sci. Transl. Med.* **6**, 224ra24 (2014).
4. Wagle, N. *et al.* High-throughput detection of actionable genomic alterations in clinical tumor samples by targeted, massively parallel sequencing. *Cancer Discov.* **2**, 82–93 (2012).
5. Heitzer, E. *et al.* Tumor-associated copy number changes in the circulation of patients with prostate cancer identified through whole-genome sequencing. *Genome Med.* **5**, 30 (2013).
6. Gertz, M. A. *et al.* Clinical implications of t(11;14)(q13;q32), t(4;14)(p16.3;q32), and -17p13 in myeloma patients treated with high-dose therapy. *Blood* **106**, 2837–2840 (2005).
7. Kumar, S. K. *et al.* Improved survival in multiple myeloma and the impact of novel therapies. *Blood* **111**, 2516–2520 (2008).
8. Chapman, M. A. *et al.* Initial genome sequencing and analysis of multiple myeloma. *Nature* **471**, 467–472 (2011).
9. Lohr, J. G. *et al.* Widespread genetic heterogeneity in multiple myeloma: implications for targeted therapy. *Cancer Cell* **25**, 91–101 (2014).
10. Schmitt, M. W. *et al.* Detection of ultra-rare mutations by next-generation sequencing. *Proc. Natl. Acad. Sci. U. S. A.* **109**, 14508–14513 (2012).
11. Kennedy, S. R. *et al.* Detecting ultralow-frequency mutations by Duplex Sequencing. *Nat. Protoc.* **9**, 2586–2606 (2014).
12. Leek, J. T. *et al.* Tackling the widespread and critical impact of batch effects in high-throughput data. *Nat. Rev. Genet.* **11**, 733–739 (2010).
13. Iglewicz, B. & Hoaglin, D. C. *How to Detect and Handle Outliers.* **16**, (ASQC Quality Press, 1993).
14. Chung, N. *et al.* Median absolute deviation to improve hit selection for genome-scale RNAi screens. *J. Biomol. Screen.* **13**, 149–158 (2008).
15. Li, H. & Durbin, R. Fast and accurate short read alignment with Burrows-Wheeler transform. *Bioinforma. Oxf. Engl.* **25**, 1754–1760 (2009).
16. McKenna, A. *et al.* The Genome Analysis Toolkit: a MapReduce framework for analyzing next-generation DNA sequencing data. *Genome Res.* **20**, 1297–1303 (2010).
17. Newman, A. M. *et al.* An ultrasensitive method for quantitating circulating tumor DNA with broad patient coverage. *Nat. Med.* **20**, 548–554 (2014).

REVIEWERS' COMMENTS:

Reviewer #3 (Remarks to the Author):

The authors have done a very thorough job in addressing the reviewers comments. Although I still have two concerns, they are not material and should not prevent publication.(1. I believe that the chosen method for mutation detection using MuTect is sub optimal and not suitable for routine use. But the authors are transparent about the use of the system, which allows the interested reader to come to their own conclusions. 2. The observed batch effect is something that needs to be better understood before routine use in clinical practice).

Congratulations on a well written manuscript.

Reviewer #6 (Remarks to the Author):

The submitted manuscript by Kis and colleagues is well written and increases the knowledge in the growing field of cfDNA testing in patients with known cancers. The authors do a great job of introducing the challenges associated with repeat bone marrow biopsy and how cfDNA testing has the potential for improving their clinical care. The authors describe the use of a small hybrid capture panel designed to interrogate the protein coding regions from 5 genes important in MM. They also describe the analytical method used and, importantly, provide access to the scripts that were used in the process of variant filtering. While this study is both scientifically interesting and potentially clinically important in the field of MM, additional commentary/clarification should be provided prior to publication.

Major comments

- The authors describe that mutations in KRAS, NRAS, and BRAF occur in 27%, 24%, and 4% of MM cases, respectively. These values do not seem to be in line with the results of this study where KRAS and BRAF positive rates seem to be far higher than these numbers. Conversely, this also suggests that there are still a large number of MM patients where additional driver mutations are present. Collectively, this point should be addressed as either a limitation or explained in the discussion section of the revised manuscript.

- The use of the term 'sensitivity' when comparing results to an alternative test that also contains uncertainties (i.e. comparing cfDNA to BM profiling) is likely overstated. A more accurate statement would be the use of the term 'concordance' when discussing the mutations/variants detected between the two tests.

- While the authors should be commended for identifying the observed batch effect and implementing a solution that seems reasonable within this study, caution should be used in over-extending the results to a clinical lab without further testing/validation. For this reason, the text on lines 146-147 of the manuscript should be softened to something like '....in data quality may be beneficial for using this method in other research laboratories and in a clinical setting, but requires additional testing and validation prior to implementation.'

- In the description of 'serial testing of patients treated with trametinib', there does not seem to be a clear trend of changing allele frequencies that aligns with response to therapy. While this could be due to a number of reasons (advanced stage at enrollment, tumor heterogeneity, etc.), the reader may be interested in hearing the authors speculate on this apparent lack of concordance in the discussion section of their manuscript.

- In the last paragraph of page 14 and throughout other portions of the revised manuscript, the authors postulate on the potential expansion of their panel without mentioning that, in addition to impacting sensitivity as they described, there is also a very likely impact on reduced specificity through panel expansion. Since their method relies upon an empirically derived distribution, the more data points that are added to the distribution, the more likely that a 'false positive' will crossover their established cutoff. This needs to be understood and addressed in this section.

- In the paragraph starting on line 275, the authors describe the novel aspects of their protocol. They state that the use of targeting all protein-coding exons of genes is novel; however, multiple

other assays describe the use of full exon sequencing of multiple genes, in particular those from commercial laboratories (Guardant Health, Personal Genome Diagnostics, Foundation Medicine, etc.).

- On line 388 of their revised manuscript, the authors state that 'this approach allowed us to effectively differentiate between germline SNPs and tumor-derived somatic variants without matching normal DNA' without providing any support that this method was 'effective.' This is particularly important in the validation cohort where multiple 'likely somatic' variants had LOD scores higher than purported SNPs. Understanding the limitation of retrospectively obtaining material for germline testing, the authors should more strongly state the need for matched normal material moving forward.

- Without performing a statistical analysis, the allele frequencies in the validation data set seem higher than those observed in the test set. If this is true, this could inflate the level of positive concordance between BM and cfDNA. Was this evaluated with the understanding that it cannot be accurately controlled in blinded samples?

- The authors should be commended for their work toward estimating the complexity of the library without single molecule barcodes. While they have included many of the crucial steps in the process whereby complexity may be reduced, I do not see an estimate in their model for the efficiency of target capture. Since they use only limited PCR cycling prior to target capture and the target capture process is known to be inefficient, this process could also result in the loss of DNA templates. The authors should also incorporate this into their model and include the results in the revised version of their supplementary methods.

Minor comments

- In the abstract, the authors refer to 'concentrations as low as 0.25%.' I feel that this would be more accurately referred to as 'allele frequencies as low as 0.25%' while also mentioning that the median frequency was much higher.

- Page 5, line 57 may read more clearly by changing 'generated using 5-gene panel' to 'generated using our 5-gene panel.'

- The authors should further elaborate on how they defined the sequencing data obtained from the three samples that failed BM testing as 'reliable.'

- The authors should reference where the comparative solid tumor cfDNA concentration data was obtained in the results section of the manuscript (Page 5, line 62).

- In the discussion section, the authors use two different numbers for specificity (predicted 100% specificity and >98% specificity). These should be consistent.

- On reference 15, the journal 'Int. J. Cancer' is repeated twice.

Response to Reviewers' Comments

Response to Reviewer #3

Reviewer's Comments: The authors have done a very thorough job in addressing the reviewers comments. Although I still have two concerns, they are not material and should not prevent publication. 1). I believe that the chosen method for mutation detection using MuTect is sub optimal and not suitable for routine use. But the authors are transparent about the use of the system, which allows the interested reader to come to their own conclusions. 2). The observed batch effect is something that needs to be better understood before routine use in clinical practice. Congratulations on a well written manuscript.

We thank the reviewer for the positive feedback and a thorough review of our manuscript.

Response to Reviewer #6

Reviewer's Comments: The submitted manuscript by Kis and colleagues is well written and increases the knowledge in the growing field of cfDNA testing in patients with known cancers. The authors do a great job of introducing the challenges associated with repeat bone marrow biopsy and how cfDNA testing has the potential for improving their clinical care. The authors describe the use of a small hybrid capture panel designed to interrogate the protein coding regions from 5 genes important in MM. They also describe the analytical method used and, importantly, provide access to the scripts that were used in the process of variant filtering. While this study is both scientifically interesting and potentially clinically important in the field of MM, additional commentary/clarification should be provided prior to publication.

Major comments

1. *The authors describe that mutations in KRAS, NRAS, and BRAF occur in 27%, 24%, and 4% of MM cases, respectively. These values do not seem to be in line with the results of this study where KRAS and BRAF positively rates seem to be far higher than these numbers. Conversely, this also suggests that there are still a large number of MM patients where additional driver mutations are present. Collectively, this point should be addressed as either a limitation or explained in the discussion section of the revised manuscript.*

We agree with the reviewer that these differences in the prevalence of *KRAS* and *BRAF* mutations should be addressed in the discussion section of our manuscript. We believe that the increased rate of detection of *KRAS*, *NRAS* and *BRAF* mutations in our study is likely due to greater sensitivity of our method for detecting low allele frequency mutations in these genes and the differences between our patient cohorts.

We have added a paragraph in the Discussion section of the revised manuscript addressing this issue:

“The prevalence of *KRAS*, *NRAS* and *BRAF* mutations in our patient cohort (38%, 23%, and 11%, respectively) was higher compared to previously published data (23-26%, 20-24%, and 4-6% of MM cases, respectively) obtained from shallow genome (30X) and whole exome (100X) sequencing studies in MM patients

(Chapman et al., 2011; Lohr et al., 2014). These differences may be explained by increased sensitivity of our method, since 37 of 48 BM tumour DNA samples analyzed in this study were sequenced using the 5-gene targeted sequencing approach (>5,000X mean target coverage). As a result, we were able to detect many mutations with AFs < 10%, including 10 of 22 *KRAS* mutations, 2 of 13 *NRAS* mutations, and 2 of 7 *BRAF* mutations that most likely would be missed by shallow genome/whole exome sequencing. In addition, treated MM patients are reported to have significantly higher AFs for somatic mutations in recurrently mutated genes such as *KRAS*, *NRAS*, and *BRAF* when compared to untreated patients (Lohr et al., 2014). This may lead to improved rate of detection of somatic variants in these genes in treated MM patients, who make up 83% of our study cohort but only 50% of the cohort in studies reporting lower prevalence of somatic mutations in these genes (Chapman et al., 2011; Lohr et al., 2014).” (Pages 13-14)

We also revised the statement in the Introduction section to include the prevalence of *KRAS*, *NRAS*, and *BRAF* mutations reported by both of the cited studies:

“Activating mutations in *KRAS*, *NRAS*, and *BRAF* genes that encode proteins with a key role in the mitogen-activated protein kinase (MAPK) pathway have been reported in 23-26%, 20-24%, and 4-6% of MM cases, respectively (Chapman et al., 2011; Lohr et al., 2014).” (Page 3)

2. *The use of the term ‘sensitivity’ when comparing results to an alternative test that also contains uncertainties (i.e. comparing cfDNA to BM profiling) is likely overstated. A more accurate statement would be the use of the term ‘concordance’ when discussing the mutations/variants detected between the two tests.*

We thank the reviewer for this recommendation and have corrected / clarified the statements made about the “sensitivity” of our test throughout the manuscript and supplementary methods.

3. *While the authors should be commended for identifying the observed batch effect and implementing a solution that seems reasonable within this study, caution should be used in over-extending the results to a clinical lab without further testing/validation. For this reason, the text on lines 146-147 of the manuscript should be softened to something like ‘...in data quality may be beneficial for using this method in other research laboratories and in a clinical setting, but requires additional testing and validation prior to implementation.’*

We thank the reviewer for this recommendation and have made the suggested revisions in the Discussion section of the revised manuscript.

“This resilience against batch effect and variation in data quality may be beneficial for using this method in other research laboratories and in clinical setting, but requires additional testing and validation prior to implementation.” (Pages 9-10)

4. *In the description of ‘serial testing of patients treated with trametinib’, there does not seem to be a clear trend of changing allele frequencies that aligns with response to therapy. While this could be due to a number of reasons (advanced stage at enrolment, tumour heterogeneity, etc.), the reader may be interested in hearing the authors speculate on this apparent lack of concordance in the discussion section of their manuscript.*

This is an important finding from the small subset of MM patients enrolled in trametinib clinical trial, which grants more attention in the main text of the paper.

In the revised manuscript, we have clarified the Results section describing these findings:

“Seven patients for which serial sampling was available were enrolled in the PHL-9460 clinical trial of MEK inhibitor trametinib for relapsed/refractory myeloma with at least two prior lines of therapy (NCT01989598). Of these seven patients, two patients (MYL-033 and MYL-049) did not have *KRAS*, *NRAS*, or *BRAF* mutations and another two patients (MYL-012 and MYL-022) had samples collected prior to and at study entry, but no serial cfDNA samples were available after starting trametinib. Only three patients had *KRAS*, *NRAS*, or *BRAF* mutations, which we were able to follow in cfDNA and BM derived tumour DNA during the course of trametinib therapy to evaluate the subclonal response to treatment. Two of these patients progressed with increasing kappa light chains and corresponding increases in AFs of *NRAS* p.Q61K and *KRAS* p.A146T (MYL-003 and MYL-018, respectively). A third patient (MYL-043) progressed on trametinib with increasing kappa light chain concentration and bony and extramedullary disease but decreasing AFs of *KRAS* p.G13D in cfDNA. When AKT inhibitor GSK2141795 was added, we observed discordant response with decrease in kappa light chain concentration and allele fractions of *KRAS* p.G13D in cfDNA and BM, but increase in M-protein (indicative of progression) and growth of a new extramedullary lesion. One of the patients without *KRAS*, *NRAS*, or *BRAF* mutations, MYL-049, progressed on trametinib with explosive extramedullary disease. We detected only *PIK3CA* mutations in this patient, including p.H59P mutation (AFs in cfDNA and BM decreased marginally during progression) and subclonal p.E545K mutation only detected in cfDNA (AF increased from 1.0% to 2.1% during progression). The other biomarker-negative patient (MYL-033) had stable disease on study with no mutations detected at screening or follow-up.” (Pages 11-12)

In addition, as recommended by the reviewer, we have included a paragraph in the Discussion section providing some insight about the observed lack of concordance between molecular and clinical measurements of treatment response in this patient:

“We analyzed serial cfDNA samples from patients enrolled in trametinib trial during the course of therapy to investigate the role of MAPK pathway mutations in patient response to trametinib treatment. In three patients with evidence of *KRAS*, *NRAS*, or *BRAF* mutations and serial plasma samples available during the course of trametinib treatment, we observed a good correlation between data from sequencing of cfDNA and matching BM-derived tumour DNA but some inconsistencies between clinical and molecular response to treatment. In two patients, clinical disease progression was associated with an increase in AFs of *NRAS* and *KRAS* mutations, in agreement with our prediction of a MAPK pathway-dependent mechanism of resistance to trametinib. However, the third patient demonstrated clinical progression, with increase in M-protein and growth of a new extramedullary lesion, but a decrease in AFs of *KRAS* p.G13D mutation in BM and cfDNA. Relatively low AF in the BM tumour DNA (19%) suggests that *KRAS* p.G13D mutation in this patient is likely subclonal. Hence, we suspect that the observed clinical progression may be caused by a differential response to treatment within the subclones, with effective trametinib response in *KRAS*-mutated subclone but not the non-mutated subclones, or appearance of new secondary resistance mutations in genes we did not examine using our 5-gene

panel analysis. To confirm our hypothesis, this question needs to be further investigated using a more comprehensive genetic profiling of MM tumour DNA and cfDNA in serial samples from a larger patient cohort. Nonetheless, our current data indicate that cfDNA sequencing may serve as a surrogate for serial sampling of the BM for monitoring patients on trial.” (Pages 14-15)

5. *In the last paragraph of page 14 and throughout other portions of the revised manuscript, the authors postulate on the potential expansion of their panel without mentioning that, in addition to impacting sensitivity as they described, there is also a very likely impact on reduced specificity through panel expansion. Since their method relies upon an empirically derived distribution, the more data points that are added to the distribution, the more likely that a ‘false positive’ will crossover their established cutoff. This needs to be understood and addressed in this section.*

This is a valid concern that requires more detailed discussion in the paper. While we agree with the reviewer that with increasing target region the number of technical errors will also increase, we predict that this will consequently change the distribution of muTect LOD scores within each sample and thereby modify the sample specific threshold for calling true variants. In fact, larger panels offer a larger genomic footprint for establishing this background error rate and may offer a more robust determination of this threshold for each sample. Ultimately, when changing gene targets or increasing the size of the target panel, an ROC curve should be used to empirically determine the appropriate modified Z score threshold for distinguishing real mutation calls from sequencing artifacts in cfDNA data generated using the new panel (the approach described in our study) to optimize sensitivity and specificity of the method. We now address this in more detail in the Discussion section:

“With increasing panel size, the sensitivity and specificity of this approach may be affected by poor mapping quality in specific genomic regions (e.g., pseudogenes, high homology regions), poor capture efficiency or coverage uniformity, increased sequence bias, and increased cost of sequencing necessary to achieve a similar depth of coverage. Some of these challenges may be overcome by parallel sequencing of matched normal DNA and/or cfDNA from healthy volunteers as well as further improvement of library preparation and target capture methodology to allow identification and removal of systematic mapping artifacts and polymerase or sequencer errors (e.g., applying duplex sequencing method with molecular barcoding and error suppression strategies)^{16,24,25}. Furthermore, to optimize the sensitivity and specificity of LB-Seq with a new target capture panel design, the appropriate modified Z-score threshold for differentiating between real mutation calls and sequencing artifacts should be determined empirically through the use of the ROC curve and training and validation cohorts, as described in this study.” (Pages 18-19)

6. *In the paragraph starting on line 275, the authors describe the novel aspects of their protocol. They state that the use of targeting all protein-coding exons of genes is novel; however, multiple other assays describe the use of full exon sequencing of multiple genes, in particular those from commercial laboratories (Guardant Health, Personal Genome Diagnostics, Foundation Medicine, etc.).*

We thank the reviewer for bringing this to our attention and have softened the language used to describe the novelty of our approach as follows:

“Our method differs from previous protocols in several important aspects. Rather than using a common approach of looking only at mutation hotspots, we targeted all protein-coding exons of genes of interest to allow in a single assay, simultaneous detection and tracking of any mutations in the target region over the course of therapy (Heitzer et al., 2013; Wagle et al., 2012).” (Page 17)

7. *On line 388 of their revised manuscript, the authors state that ‘this approach allowed us to effectively differentiate between germline SNPs and tumour-derived somatic variants without matching normal DNA’ without providing any support that this method was ‘effective.’ This is particularly important in the validation cohort where multiple ‘likely somatic’ variants had LOD scores higher than purported SNPs. Understanding the limitation of retrospectively obtaining material for germline testing, the authors should more strongly state the need for matched normal material moving forward.*

We agree with the reviewer that the need for germline testing to improve the identification of germline SNPs in cfDNA data should be emphasized in the paper and have revised the statement in the Methods section accordingly:

“To ensure accurate identification of germline variants, we recommend sequencing matched normal genomic DNA obtained from leukocytes readily available from the same blood samples used for cfDNA extraction, especially in patients with evidence of high-AF mutation calls (>40%) or when applying this method to a larger target panel design.” (Pages 23-24)

8. *Without performing a statistical analysis, the allele frequencies in the validation data set seem higher than those observed in the test set. If this is true, this could inflate the level of positive concordance between BM and cfDNA. Was this evaluated with the understanding that it cannot be accurately controlled in blinded samples?*

This is a valid concern raised by the reviewer. When designing this research study, we chose to be blinded on the selection of study participants and processed and sequenced all cfDNA and BM samples in the order as they were transferred to our laboratory from our clinical team. Patient samples were grouped into training or validation cohort based only on the timing of sample collection and sequencing (first 25 cfDNA samples with matching BM data available were used in the training cohort, all remaining samples were included in the validation cohort). We performed statistical analysis (Wilcoxon rank sum test) and the allele fractions detected in the training cohort were not statistically different from those observed in the validation cohort for cfDNA ($P = 0.083$, Supplementary Fig. 12a) or BM ($P = 0.073$, Supplementary Fig. 12b) sequencing data, as shown below. The observed trend towards higher allele fractions in the validation cohort may be explained by higher proportion of untreated patients in the training cohort (4/23, 17%) compared to the validation cohort (2/17, 12%), in agreement with other studies (Lohr et al., 2014).

We now address this issue in the Results section of the revised manuscript:

“Compared to training cohort, we observed a trend towards higher mutant allele fractions in samples included in the validation cohort; however, this difference was not significant ($P = 0.083$ and 0.073 for cfDNA and BM data in **Supplementary Fig. 12a,b**, respectively).” (Page 9)

Supplementary Figure 12. Comparison of mutant allele frequencies in training and validation cohorts. The distributions of mutant allele frequencies detected in cfDNA (**a**) or BM tumour DNA (**b**) sequencing data in training and validation cohort samples are shown as box plots, where the central rectangle spans the first to the third quartile (interquartile range or IQR). A segment inside the rectangle shows the median, and "whiskers" above and below the box show the value $1.5 \times \text{IQR}$ above or below the third or the first quartile, respectively. The number of mutations included in each analysis is shown under group name in the figure. Wilcoxon signed-rank test was used for comparison of two groups in each panel with P value of 0.05 considered statistically significant.

9. *The authors should be commended for their work toward estimating the complexity of the library without single molecule barcodes. While they have included many of the crucial steps in the process whereby complexity may be reduced, I do not see an estimate in their model for the efficiency of target capture. Since they use only limited PCR cycling prior to target capture and the target capture process is known to be inefficient, this process could also result in the loss of DNA templates. The authors should also incorporate this into their model and include the results in the revised version of their supplementary methods.*

We thank the reviewer for the positive feedback about our effort to estimate the complexity of our cfDNA libraries and appreciate the suggestion to include target capture efficiency in our model. Given the redundancy of each unique DNA template present in the library pool (after 4-8 cycles of PCR amplification), we believe that the inefficiencies during hybridization and target capture alone are unlikely to lead to significant reduction in library complexity. Each cycle of PCR amplification of captured DNA (13-15 cycles) will

introduce additional loss of DNA material as well as potential amplification bias. Given the small size of our target capture panel (<18 kb, which makes up only 0.0006% of the genome), we only quantified the final amplified and purified DNA product prior to sequencing. Intermediate DNA quantification after target capture but prior to amplification is very challenging and would result in substantial loss of captured DNA material. Hence, in this study we were unable to estimate the proportion of DNA molecules lost during each step involved in target capture protocol going from pooled DNA libraries to the final captured amplified DNA.

With these limitations in mind, we have taken several measures to improve our target capture efficiency and have described these in detail in the Supplementary Methods section (Page 36-38, shown below). We also refer readers to a recent study by Newman et al. (co-authored by Dr. Scott Bratman, one of the co-authors of this manuscript) that used molecular barcode-ligated cfDNA libraries and the mark and recapture method to estimate that “~50-60% of total human genome equivalents that entered the library preparation process made it through post-capture PCR” (Newman et al., 2016). Although the methodology used in our study was different (e.g., using sample barcodes and a different target capture panel) we anticipate our rates of recovery of unique DNA fragments during sample processing to be similar to these published data.

We have revised the Results section as follows:

“Although target capture is also associated with additional loss of library fragments due to the inefficient hybridization of target probes to their target DNA fragments and loss of hybridized DNA during streptavidin bead capture, the redundancy of library molecules available for capture should prevent significant reduction in library complexity during target capture process. In this study, we were unable to measure the efficiency of target capture; however, a previous study by Newman et al. estimated using molecular barcoding and mark and recapture method that approximately 50-60% of unique cfDNA fragments that enter into the library preparation process are represented in the final sequencing data after target capture and PCR (Newman et al., 2016).” (Page 6-7)

We also revised the corresponding section of the Supplementary Methods:

“Estimation of duplication rate and loss of unique cfDNA molecules during sample processing

For a typical cfDNA sample, 83 ng of cfDNA was used to prepare adapter-ligated sequencing library. Since cfDNA is naturally fragmented with a major fragment size corresponding to a DNA loop around one nucleosome (~166 bp, 80-85%) and a second peak (~15-20%) corresponding to two nucleosomes (332 bp), we estimated the average fragment size of cfDNA to be approximately 180 bp. Using 650 Daltons as an average weight of one DNA base pair, we estimated that 83 ng of cfDNA contains approximately 4.3×10^{11} unique DNA fragments (7.1×10^{-13} mol). For libraries prepared using 83 ng of input cfDNA (n=33), the library recovery after ligation to NEXTflex-96 adapters and four amplification cycles was median 43% of the theoretical maximum yield (21-70% range, **Supplementary Fig. 3b**). Some of this reduction in yield is due to loss of unique DNA templates during adapter ligation and initial bead clean-up, and some is likely caused by inefficiencies in post-ligation PCR and final library clean-up, primarily causing loss of redundant library fragments without major effect on library complexity. If we

presume that the entire reduction in library yield is due to loss of unique DNA templates (the most conservative estimate of library complexity), the estimated number of unique cfDNA molecules converted into a barcoded DNA-seq library is 1.8×10^{11} ($0.88\text{--}3.0 \times 10^{11}$). Since the target region (146×120 bp probes = 17.5 kbp) makes up only 0.0006% of the genome, we estimated that a similar proportion of cfDNA molecules present in each sample will align to the region of interest and will be captured for sequencing (1.1×10^6 unique cfDNA fragments). Since ligated fragments within the library are amplified 4 PCR cycles (theoretically resulting in 16 copies of each molecule within the pool), the downstream steps are less likely to result in complete allele drop out, despite known inefficiencies in hybridization of target probes to DNA fragments, capture of hybridized DNA probes, post-capture PCR, or magnetic bead clean-up of amplified DNA product. Not all cfDNA fragments containing target DNA sequence will bind to their complementary DNA-binding probes, some of the hybridized DNA will not be captured by streptavidin beads or will be lost during clean-up, and each cycle of PCR amplification after target capture will introduce additional loss of DNA material as well as potential amplification bias.

Given the small size of our target capture panel (17.5 kb), we only quantified the final amplified and purified DNA product prior to sequencing, but did not quantify DNA yield immediately after target capture but prior to amplification, which would result in substantial loss of captured DNA material. Without this intermediate quantification or the use of molecular barcoding strategies (Kennedy et al., 2014; Newman et al., 2016; Schmitt et al., 2012), it is difficult to estimate the proportion of DNA molecules lost during each step involved in target capture protocol.

In light of these limitations, we have taken several measures to improve our target capture efficiency such as i) using individually synthesized high quality DNA binding probes (designed, synthesized, and individually tested by the Integrated DNA Technologies); ii) using an abundance of target capture probes (0.002 μ mol of each probe) per target capture reaction; and iii) increasing the hybridization time from 4 hours to 16-20 hours (overnight) at 65°C in order to improve probe hybridization to target DNA. Our protocol also allows direct PCR amplification of captured DNA without intermediate DNA clean-up, with streptavidin beads left in solution along with captured DNA templates during 13-15 cycles of PCR amplification, which reduces the loss of captured DNA templates. For all PCR steps in our protocol, we used KAPA HiFi polymerase high fidelity enzyme, which has been shown to improve amplification efficiency and reduce amplification bias for extremely GC-rich or AT-rich fragments (Quail et al., 2011). Together with the redundancy of each unique DNA template present in the library pool, we believe that the inefficiencies during hybridization, target capture, and post-capture PCR are unlikely to lead to significant reduction in library complexity.

A study by Newman et al. used molecular barcode-ligated cfDNA libraries and the mark and recapture method to estimate that approximately 50-60% of total human genome equivalents which enter into the library preparation are present after target capture and post-capture PCR (~40-50% loss of unique cfDNA templates) (Newman et al., 2016). Although the methodology used in our study was different (e.g., using sample barcodes and a different target capture panel), we anticipate our rates of recovery of unique cfDNA fragments during sample processing to be similar to these published data.

Assuming 180 bp to be the average size of cfDNA fragments, approximately 98 fragments will be required to achieve 1X coverage of the 17.5 kbp target region. Using our conservative estimates that 1.1×10^6 unique cfDNA fragments will align to the target region and will be captured for sequencing, we estimate that the median depth of coverage that can be achieved through sequencing of 1.1×10^6 unique cfDNA fragments is approximately 11,000X. Hence, mutations detected in a region with 20,000X coverage should have ~55% unique reads, and this fraction will decrease in regions with higher coverage (~11% unique reads for regions with 100,000X coverage). Higher degree of duplication is likely to be present in samples with lower initial cfDNA input into library constructions or in libraries prepared using Illumina LT adapters (the ligation efficiency was lower compared to NEXTflex adapters, **Supplementary Fig. 3**). If the duplication rate is similar between normal and tumour-derived cfDNA fragments during sample processing, the AFs observed for tumour-derived mutations remain largely independent of the duplication rate and representative of the allele fractions present in plasma.” (Pages 37-38, Supplementary Information)

Minor comments

1. *In the abstract, the authors refer to ‘concentrations as low as 0.25%.’ I feel that this would be more accurately referred to as ‘allele frequencies as low as 0.25%’ while also mentioning that the median frequency was much higher.*

We thank the reviewer for this recommendation and have revised the abstract as follows:

“This method includes custom variant filtering algorithms that enabled detection of tumour-derived fragments present at allele frequencies as low as 0.25% in cfDNA (11% average, 0.25-46% range).” (Page 2)

2. *Page 5, line 57 may read more clearly by changing ‘generated using 5-gene panel’ to ‘generated using our 5-gene panel.’*

This is a good suggestion and we have revised the text accordingly:

“When BM sequencing data were available from more than one source (**Supplementary Table 1**), we used data generated using our 5-gene panel.” (Page 5)

3. *The authors should further elaborate on how they defined the sequencing data obtained from the three samples that failed BM testing as ‘reliable.’*

In the three BM-derived tumour DNA samples that failed clinical testing but were later sequenced in our laboratory using our 5-gene panel targeted sequencing method, the yield of CD138+ malignant cells from the BM marrow aspirate was low providing insufficient material for clinical sequencing of these samples. Our method relies on using lower amount of starting material as input for DNA sequencing library preparations and higher recovery of unique DNA molecules through this process. In all three cases we used 83 ng of sheared genomic DNA isolated from selected CD138+ tumour cells (~100 ng starting tumour DNA material sheared to ~300 bp size and quantified again post shearing). Following similar stringent protocol used for cfDNA library construction and

target capture, we were able to generate good quality targeted sequencing libraries, which passed all quality control metrics used for assessment of cfDNA sample and data quality. We achieved > 5,000X mean target coverage in all BM-derived tumour DNA samples sequenced in our laboratory with suitable enrichment of target regions and no indications of low library complexity. Based on these quality metrics, we considered these data to be reliable and included them in our manuscript.

We have clarified this in the Results section of our manuscript as follows:

“For three BM aspirate samples unsuitable for clinical grade testing due to low yield of malignant plasma cells, we were able to obtain reliable sequencing data using our 5-gene targeted deep sequencing assay because of lower requirement for input material (~100 ng of purified tumour DNA) and much higher depth of coverage (>5,000X) compared to clinical testing (~500X).” (Page 5)

4. *The authors should reference where the comparative solid tumour cfDNA concentration data was obtained in the results section of the manuscript (Page 5, line 62).*

We thank the reviewer for this suggestion and have added this information in the Results section:

“We detected higher cfDNA concentrations in this MM cohort compared to 56 patients with advanced solid tumours also processed in our laboratory using the same method (median 20.1 versus 10.3 ng per mL plasma, *P* value < 0.001, **Fig. 1a**. See **Supplementary Table 2** for cancer types).” (Page 5)

5. *In the discussion section, the authors use two different numbers for specificity (predicted 100% specificity and >98% specificity). These should be consistent.*

Thank you for bringing this to our attention. We have corrected this inconsistency and now refer to >98% specificity of this method throughout text.

6. *On reference 15, the journal ‘Int. J. Cancer’ is repeated twice.*

We thank the reviewer for identifying this error in the citation, which we fixed in the revised manuscript.

References

- Chapman, M.A., Lawrence, M.S., Keats, J.J., Cibulskis, K., Sougnez, C., Schinzel, A.C., Harview, C.L., Brunet, J.-P., Ahmann, G.J., Adli, M., et al. (2011). Initial genome sequencing and analysis of multiple myeloma. *Nature* 471, 467–472.
- Heitzer, E., Ulz, P., Belic, J., Gutsch, S., Quehenberger, F., Fischereder, K., Benezeder, T., Auer, M., Pischler, C., Mannweiler, S., et al. (2013). Tumor-associated copy number changes in the circulation of patients with prostate cancer identified through whole-genome sequencing. *Genome Med.* 5, 30.
- Kennedy, S.R., Schmitt, M.W., Fox, E.J., Kohn, B.F., Salk, J.J., Ahn, E.H., Prindle, M.J., Kuong, K.J., Shen, J.-C., Risques, R.-A., et al. (2014). Detecting ultralow-frequency mutations by Duplex Sequencing. *Nat. Protoc.* 9, 2586–2606.
- Lohr, J.G., Stojanov, P., Carter, S.L., Cruz-Gordillo, P., Lawrence, M.S., Auclair, D., Sougnez, C., Knoechel, B., Gould, J., Saksena, G., et al. (2014). Widespread genetic heterogeneity in multiple myeloma: implications for targeted therapy. *Cancer Cell* 25, 91–101.
- Newman, A.M., Lovejoy, A.F., Klass, D.M., Kurtz, D.M., Chabon, J.J., Scherer, F., Stehr, H., Liu, C.L., Bratman, S.V., Say, C., et al. (2016). Integrated digital error suppression for improved detection of circulating tumor DNA. *Nat. Biotechnol.* 34, 547–555.
- Quail, M.A., Otto, T.D., Gu, Y., Harris, S.R., Skelly, T.F., McQuillan, J.A., Swerdlow, H.P., and Oyola, S.O. (2011). Optimal enzymes for amplifying sequencing libraries. *Nat. Methods* 9, 10–11.
- Schmitt, M.W., Kennedy, S.R., Salk, J.J., Fox, E.J., Hiatt, J.B., and Loeb, L.A. (2012). Detection of ultra-rare mutations by next-generation sequencing. *Proc. Natl. Acad. Sci. U. S. A.* 109, 14508–14513.
- Wagle, N., Berger, M.F., Davis, M.J., Blumenstiel, B., Defelice, M., Pochanard, P., Ducar, M., Van Hummelen, P., Macconail, L.E., Hahn, W.C., et al. (2012). High-throughput detection of actionable genomic alterations in clinical tumor samples by targeted, massively parallel sequencing. *Cancer Discov.* 2, 82–93.